



# Automated detection of ice cliffs within supraglacial debris cover

Sam Herreid and Francesca Pelicciotti

Department of Geography, Faculty of Engineering and Environment, Northumbria University, Newcastle upon Tyne, UK

*Correspondence to:* Sam Herreid (samherreid@gmail.com)

**Abstract.** Ice cliffs within a supraglacial debris cover have been identified as a source for high ablation relative to the surrounding debris-covered area. Due to their small relative size and steep orientation, ice cliffs are difficult to detect using nadir-looking space borne sensors. The method presented here uses surface slopes calculated from digital elevation model (DEM) data to map ice cliff geometry and produce an ice cliff probability map. Surface slope thresholds, which can be sensitive to geographic

location and/or data quality, are selected automatically. The method also attempts to included area at the (often narrowing) ends of ice cliffs which could otherwise be neglected due to signal saturation in surface slope data. The method was calibrated in the Eastern Alaska Range, Alaska, USA, against a control ice cliff dataset derived from high resolution visible and thermal data. Using the same input parameter set that performed best in Alaska, the method was applied against ice cliffs manually mapped in the Khumbu Himal, Nepal. Our results suggest the method can accommodate different glaciological settings and different

DEM data sources without a data intensive (high resolution, multi-data source) re-calibration.

## 1 Introduction

Ice cliffs are steep, bare-ice surface features that can develop within a debris-covered portion of a glacier. The direct atmosphere-ice interface can result in significantly higher ablation rates relative to the surrounding debris-covered area (Buri et al., 2016b; Thompson et al., 2016; Brun et al., 2016). The mechanism(s) of ice cliff formation, the controls of ice cliff migration patterns

and ice cliff residence time on a glacier are gaining research attention but are still poorly understood processes and a present lack of base data is an obstacle to establishing a robust understanding (Reid and Brock, 2014; Watson et al., 2017). Melt and surface energy fluxes at specific ice cliffs have been studied in detail (Sakai et al., 1998; Han et al., 2010; Sakai et al., 2002; Reid and Brock, 2014; Buri et al., 2016a) and digital elevation model (DEM) differencing has shown the spatial trends of enhanced glacier melt relative to surrounding debris cover and ice cliff evolution at the scale of several cliffs or a single glacier

tongue (Thompson et al., 2016; Brun et al., 2016). All of the studies mentioned suggest that ice cliffs, if present within a debris cover, need to be accounted for in order to adequately model glacier mass change and response to climate. A wide range of ice cliff abundance within a debris covered area is possible, from no ice cliffs to an abundance capable of possibly negating, or even reversing, the melt reducing effect of surrounding debris cover (Kääb et al., 2012; Basnett et al., 2013; Gardelle et al., 2013).

To the knowledge of the authors, five methods have been used to map ice cliffs: (1) field mapping (e.g. Steiner et al., 2015); (2) manual digitization from remote sensing data (e.g. Sakai et al., 1998; Han et al., 2010; Thompson et al., 2016; Watson




et al., 2017); (3) automatically, using a surface slope threshold (e.g. Reid and Brock, 2014); (4) automatically by a principal component analysis using visible near infrared and shortwave infrared satellite bands (Racoviteanu and Williams, 2012); and (5) automatically by a object based image analysis of unmanned aerial vehicle data (Kraaijenbrink et al., 2016). None of the remote sensing studies listed offer a confidence metric based on independent data for their ice cliff map products and field

mapping is not realistic for large-scale analysis.

The objective of this paper is to present a new approach to automate the detection of ice cliffs. The method (1) requires conventional input data that are starting to become freely available globally; (2) automatically selects threshold values that can accommodate varying surface textures (e.g. glacier area with a characteristically smooth debris cover or a characteristically rough/hummocky debris cover); and (3) is assessed for quality against additional high resolution visible and thermal data.

**1.1  Formulation of the problem**

The importance of ice cliffs has become increasingly clear (Sakai et al., 1998; Han et al., 2010; Sakai et al., 2002; Watson et al., 2017), but the mapping of these features remains a challenge, especially at spatial scales beyond a few glaciers. The map-view surface expression of an ice cliff is often a crescent, circular or linear swath of steep, bare (or thinly debris covered) glacier ice surrounded by a debris layer. Steep glacier ice not completely surrounded by debris cover might exhibit melt and evolution

patterns similar to ice cliffs, but the lack of a bounding debris cover makes these areas characteristically distinct from ice cliffs and thus excluded in this study. It is unclear at the present time if a small area of low angle (i.e. not cliff) bare glacier ice enclave within a debris cover or a narrow swath of ice constrained by debris cover (e.g. a narrowing gap between two widening medial moraines) should be considered similar to ice cliffs with respect to relative melt rates in a debris-covered environment, but for this study we maintain a focus on identifying only steep features.

Watson et al. (2017) report that most ice cliffs within a subset of glaciers in the Central Himalaya are 200 m or less in length with a length of 20-40 m being the most frequent. Thompson et al. (2016) report a mean ice cliff height of 15.5 m for Ngozumpa Glacier in Nepal with notable outliers up to ~45 m. No current literature suggest other glacierized regions on Earth have ice cliffs with dimensions that deviate wildly from these localized findings. Due to this relatively small size and the high slope angle of an ice cliff, a nadir looking sensor will capture map-view ice cliff width as $D$, the distance from the bottom

debris-ice interface to the top debris-ice interface, reduced by a factor of $\int_0^D cos(\beta)dx$, where $\beta$ is surface slope along the ice cliff transect, oriented parallel to the $x$-axis. This (likely) narrow map-view area means that even in an ideal situation where there is no debris on an ice cliff face, the optically sharp boundary between rock and ice could be saturated or completely muted in remote sensing data where ice cliff area does not occupy a sufficient fraction of a data pixel. A DEM-derived surface slope expression of an ice cliff is not encumbered by debris cover on the cliff face, yet the 'true' steep slopes of an ice cliff can also

be saturated or completely muted if the spatial resolution of the computed slopes is coarse to a point where no slope value is calculated solely from pixels located within an ice cliff face. For both visible and DEM data, in the common case where an ice cliff narrows gradually at the cliff ends, ice cliff edge defining signal saturation is likely to increase towards the narrow ends and could cause a systematic underestimation of ice cliff area if left unaccounted for.





If cloud free, ablation season visible spectrum imagery is used to map ice cliffs, cliff aspect, surrounding topography and sun position at the time of data acquisition control whether the surface will be shaded or illuminated. North and south facing ice cliffs will likely be optically distinct and crescent to circular ice cliffs will likely exhibit a spectrum of shade and illumination. Automated or manual ice cliff mapping techniques using cloud free visible spectrum imagery would likely need to mitigate
this factor and devise a way to discriminate between shadowed ice cliff area and shadowed debris-covered area.

An important factor pertaining to identifying and mapping ice cliffs is the presence of thin or sparse debris cover on an ice cliff face (hereafter referred to as a "thin" debris cover, although also describing sparse debris cover that could include large clasts or boulders). A "thin" debris layer is completely undetectable from moderate resolution DEM data and can introduce ambiguities if identifying/mapping ice cliffs from visible spectrum or thermal imagery. The surface of an ice cliff face during
the melt season can vary with the deposition and/or removal of rock fragments. This process could be a slow evolution (e.g. coupled with melt, neighboring sediment distributions and englacial debris concentration) or a near instantaneous result from a local storm (e.g. windblown silt accumulations on wet, rough ice cliff surfaces). These processes lead to an ambiguity in defining an ice cliff where time may need to be considered. For example, if an image of a cliff shows that 30% of the surface area within the cliff face is comprised of large rocks caught on narrow ice ledges, should this area be excluded from what
is called an ice cliff or can it be assumed this debris cover is transient and superfluous to consider? At large scales, a time consideration of debris cover within ice cliff faces is unrealistic, yet a 30% error could have a large and compounding impact on, for example, a study calculating energy fluxes. Furthering this example to the case where over time a cliff face is 100% debris covered, there are two classification possibilities: the cessation of being an ice cliff or, if the cliff exhibits some unique signature (e.g. a thermal anomaly and/or fine sediment/clast size distribution relative to the surrounding debris cover), it could
still be considered an ice cliff.

Considering the cessation case where ice cliff area transitions to debris-covered area in the wider context of non-ice cliff yet steep debris-covered glacier area (e.g. the side of a medial moraine), it is not unrealistic that the true distributions of a population of ice cliff surface slopes and a population of debris-covered area surface slopes will have some overlap. If true, this implies that a simple surface slope threshold alone cannot cleanly identify ice cliff area.
A highly successful automated or manual ice cliff mapping technique will likely require a combination of multiple input datasets (e.g. visible and thermal data or visible and elevation data) yet still, ambiguities with defining what is and is not an ice cliff will likely remain regardless of approach.

Ultimately, the leading obstacle to successfully identifying ice cliffs is data resolution and quality. Data with a spatial resolution <1 m are ideal for mapping ice cliffs, however, at the present time, data at this resolution are often not freely
available, particularly at large scales (e.g. whole mountain ranges).

With an effort to balance ice cliff identification success rate with input data that is conventional and becoming freely available at wide spatial scales, the automated method presented here uses moderate resolution (~5 m) digital elevation data alone to identify ice cliffs. The method includes a procedure to identify ice cliff area at the ends of ice cliffs that have a narrowing end geometry. <1 m resolution visible imagery collected in the Alaska Range are used to assess the abundance of "thin" debris
cover on ice cliff faces.



## 2 Data

### 2.1 Input data

There are three required datasets for this method: (1) glacier area over which the method is applied; (2) multispectral satellite imagery; and (3) a moderate resolution DEM. Since the DEM alone is used to identify ice cliffs (the likely most temporally

transient feature considered), it is not crucial that all three data sets be coincident in time. However, debris-covered area and glacier margins should be assessed to ensure they have not changed significantly over the time span of the data used.

#### 2.1.1 Glacier area

The spatial domain over which ice cliffs are detected is bound by a user defined polygon. The perimeter can outline a portion of a glacier, a whole glacier or many glaciers and can be a mix of debris covered and debris free glacier area. A subset of the

Randolph Glacier Inventory (Pfeffer et al., 2014) is a suitable input but should be assessed for accuracy to avoid any erroneous inclusion of off glacier slopes which could be misidentified as ice cliff area and skew computed statistics. Computational cost might become a factor for typical desktop or laptop computers if solving over a large domain. This issue is addressed in Sect. 3.3.

#### 2.1.2 Satellite imagery

Multispectral satellite imagery is only used to map debris cover. The ratio of a near infrared (NIR) band and a shortwave infrared (SWIR) band is used to empirically remove radiance value variance from topographic illumination angles and, to some degree, cast shadows (Vincent, 1973). Data from the NASA/USGS Landsat program (used in this study) and ESA Sentinel-2 are two data sources that meet the input spectral and resolution requirements to map debris cover and are freely available.

#### 2.1.3 DEM

Elevation data are key in both identifying the location of ice cliffs and also defining their area. For the results of the method to be meaningful, an input DEM must have sufficient resolution and precision to resolve topography below or near the size of most ice cliffs within an area of interest. Because ice cliff locations are not being identified as the residual of DEM differencing, a high vertical accuracy is not critical. This can simplify data processing if structure from motion data are used to derive the input DEM. Photogrammetric methods are often very successful at resolving vertical precision (i.e. relative topography) and

geolocation in $x,y$ is usually simple to achieve, but a high vertical accuracy with respect to the geoid (i.e. true elevation) requires a suitable ground control point network (Westoby et al., 2012).

   DEM data that meet this criteria are not freely available for all glacierized regions on Earth at the present time. However, initiatives such as the Interagency Arctic Research Policy Committee (IARPC) Arctic DEM, which is releasing a freely available 2 to 8 m resolution DEM for all landmass above 60° and the entire State of Alaska, show promise that high resolution

DEM data may soon be available globally.





## 2.2 Calibration and validation data

The parameters of this method were calibrated using data that cover a portion of the Canwell Glacier in the Eastern Alaska Range, Alaska, USA. To test transferability, the same parameter set was applied to a portion of Ngozumpa Glacier in the Khumbu Himal, Nepal.

### 2.2.1 Canwell Glacier

Canwell Glacier (Fig. 1, 2a) is a 60 km$^2$, northwest flowing glacier in the eastern Alaska Range (63° 19.8'N, 145° 32'W). Canwell Glacier was selected for this study because several different surfaces exist in close proximity: an expansive ice cliff network in debris cover that transitions, orthogonal to flow, to bare glacier ice and a medial moraine.

On the 29th of July, 2016 between 11:00 and 11:16 local Alaska Time, nadir (or near-nadir) looking visible and thermal infrared images were collected from a helicopter over 1.7 km$^2$ of the Canwell Glacier capturing all of the different surface types listed above (Fig. 3a). The images were collected below a high overcast ceiling. This caused subtle cloud effect to be captured by varying light penetration of the cloud layer, however, this also removed the likely more negative effect of shading discussed in Sect. 1.1. 250 visible spectrum images were collected with a Canon EOS 70D camera. These images, in conjunction with 9 ground control points, were used to generate a ~8 cm resolution orthomosaic and a 1 m resolution (resampled to 5 m to match a spatial resolution more common from space borne sensors) DEM using the proprietary software Agisoft PhotoScan Professional Edition. 34 (suitable for use) thermal images were collected with a FLIR T620 camera and processed using the proprietary software FLIR Tools. Emissivity was held constant at 0.95, atmospheric temperature and relative humidity where measured from the helicopter during image acquisition using a Kestrel 4000 Weather Meter and distance from the sensor to the glacier surface was estimated using camera locations derived by Agisoft Photoscan. The thermal images were manually georeferenced to match the orthomosaic image described above.

A nearly cloud-free Landsat 8 image was acquired over the Canwell Glacier (path/row: 67/16) on the 31st of August, 2016. Debris cover extent and glacier margins were found to be effectively static over the 33 day interval between the acquisition of helicopter-borne and Landsat 8 data.

### 2.2.2 Ngozumpa Glacier

Ngozumpa Glacier (Fig. 1, 2b) is a 60 km$^2$ south flowing glacier in Khumbu Himal (27° 57'N, 85° 42'E). Ngozumpa Glacier was selected for this study for two reasons, first, it is located in a distinctly different geographical location with a debris cover that is typical of the Himalaya and notably different from the Canwell Glacier. Fig. 3 illustrates some differences between Canwell and Ngozumpa glaciers including rock/boulder size, ice cliff size, amount of "thin" debris cover on ice cliff faces and overall hummocky nature of the debris-covered area. The debris on Ngozumpa Glacier is thick (1-3 m towards the terminus (Nicholson, 2005)) and covers the full width of the glacier continuously for nearly the entire ablation zone. Differences in the two inset histograms in Fig. 3 suggest even if overlooking overlap errors, a simple surface slope threshold deemed suitable to define ice cliffs at one location cannot be assumed to have wide reaching applicability. The second reason Ngozumpa Glacier





was selected is that an automated ice cliff map can be assessed against the manually generated ice cliff map from Thompson et al. (2016), allowing the removal of some potential manual delineation bias in this study.

Thompson et al. (2016) provided their GeoEye-1 orthoimage acquired on the 29th of December, 2012, the corresponding stereo image derived DEM (1 m resolution, resampled to 5 m for this study) and their ice cliff map generated manually using both the orthoimage and DEM. Additionally, Thompson et al. (2016) generated and provided a mask of area where surface elevation was poorly resolved. The method Thompson et al. (2016) used to map ice cliffs was to define a line along the top edge of each ice cliff based on optical characteristics and steep surface slopes calculated from the DEM. In order to conduct a quality assessment between the top edge lines defined by Thompson et al. (2016) and the automated ice cliff polygons identified using the method presented here, the ice cliff top edges from Thompson et al. (2016) were manually adjusted to polygons incorporating the area of each ice cliff using the December 23rd, 2012 visible GeoEye-1 imagery. Some smaller ice cliff additions were also made.

A nearly cloud-free Landsat 8 image was acquired over Ngozumpa Glacier (path/row: 140/41) on the 30th of November, 2014. Because the portion of Ngozumpa Glacier considered in this study is 100% debris covered, a debris extent map generated with a 2 year gap does not alter the spatial domain. The glacier margin was mapped from the December 23rd, 2012 GeoEye-1 imagery.

## 3 Methods

### 3.1 Isolation of debris-covered area

The spatial domain is refined from total glacierized area, including bare ice and accumulation zone area, to only debris-covered glacier area. Debris-free area is identified and removed using the band ratio of NIR and SWIR satellite bands with a user defined threshold (Paul et al., 2004)(Table 1). To prevent the removal of bare ice pixels that are part of an ice cliff, closed shapes identified as bare ice within the debris-covered area are filled (reclassified as debris-covered area) if below a user defined threshold area (Table 1). This is possible where an ice cliff is debris-free and big enough to cause one or more satellite image pixels to fall below the NIR/SWIR ratio threshold. Bare ice below the area threshold (included as debris-covered area) that is not part of an ice cliff will be rejected as ice cliff area in the subsequent step based on low surface slopes.

### 3.2 Ice cliff identification

#### 3.2.1 Iterative ice cliff detection

Within debris-covered area, surface slope ($\beta$) is calculated as the maximum rate of change in elevation for each DEM pixel value relative to its 8 neighbor values. A threshold slope value ,$\beta_i$, isolates steeper area from which statistics and threshold values are derived. The tool is run iteratively, varying $\beta_i$ over the full range of possible surface slopes (below over-vertical), from 0 to 90° in $n$ iterations ($i$).



For each $i$, two areas are defined that will, together, define area with a high likelihood of being an ice cliff: (1) an initial ice cliff area ($A_i$) from which statistics are computed, further geometries are derived and the base shape of the final ice cliff area is defined; and (2) $A_{ei}$, an area slightly more encompassing than $A_i$ from which lengthwise ends of the final ice cliff geometries are extracted (subscript '$e$' for '$end$').

5      $A_i$ is defined as $area(\beta > \beta^*_i)$ where $\beta^*_i = mean(\beta > \beta_i)$ (Fig. 4a). Using $\beta^*_i$ rather than simply $\beta_i$ speeds up computation by discarding the case where ice cliffs occupy an overwhelming percentage (>>50%) of the debris-covered area. If ice cliffs did occupy one half or more of a debris-covered area (in map-view), its classification as a debris-covered portion of a glacier could be questioned.

Ice cliff centerlines are computed by creating a Voronoi cell for each vertex in the outline of $A_i$ converted to a dense set 10   of vertices. The bounding edges of each Voronoi cell is removed except for the edge in the center of a shape in $A_i$. A point removal line simplification is applied to smooth extraneous bends, particularly at centerline ends. The centerline ends are then extended by a user defined distance, $L_e$, with the topological restriction that centerline extensions can intersect, but not cross one another (Fig. 4b). The extended centerlines are then transformed to an area, $B_{ei}$, by a buffer distance, $\alpha$, applied outward in all directions (Fig. 4b). $C_{ei}$ is the intersection of $B_{ei}$ and $A_{ei}$, where $A_{ei}$ is area with a surface slope greater than $\beta^*_i$ relaxed 15   by a user defined factor, $\beta_e$ (Fig. 4c). $C_{ei}$ is intended to identify area that is part of an ice cliff but expressed by surface slopes less than $\beta^*_i$ due, possibly, to narrowing ice cliff ends where DEM data and subsequent surface slope calculations saturate a true, steep surface slope signal. Area with a high likelihood of being an ice cliff, $A_{cliff_i}$, is defined as the union of $C_{ei}$ and $A_i$ where a user defined minimum shape area threshold, $A_{min}$ is exceeded (Fig. 4d).

$A_{cliff_i}$ is the definitive ice cliff area used for the error analysis in Sect. 3.5 after optimization described in Sect. 3.2.2, but 20   for some applications of this method, a distributed probability map might be a more useful product. With $\beta_i$ as a lower limit and $\beta_u$, where $\beta_u = \beta^*_i + std(\beta > \beta_i)$, as an upper limit, a piecewise probability model can be defined as

$$p(x \mid \beta, \omega) = \begin{cases} 0, & \beta < \beta_i,\ \omega = \{Y, N\} \\ \dfrac{\beta + \beta_i}{\beta_u - \beta_i}, & \beta_i \leq \beta \leq \beta_u,\ \omega = Y \\ \dfrac{\varphi(\beta + \beta_i)}{\beta_u - \beta_i}, & \beta_i \leq \beta \leq \beta_u,\ \omega = N \\ \varphi, & \beta > \beta_u,\ \omega = N \\ 1, & \beta > \beta_u,\ \omega = Y \end{cases} . \tag{1}$$

Where the probability of each pixel being part of a true ice cliff, $x$, is assigned given surface slope, $\beta$, and $\omega$, where $\omega$ is a binary classification $\{Y, N\}$ for pixels falling within $A_{cliff_i}$, $Y$, and outside $A_{cliff_i}$, $N$. For pixels where $\omega = N$, ice cliff 25   liklihood is reduced by a user defined factor, $\varphi$. $\varphi = 0$ implies the iterative process will have zero error, $\varphi = 1$ discards the entire iterative process and $\varphi = 0.5$ (for example) means that a surface slope $> \beta_u$ but not bound by a high likelihood ice cliff shape ($A_{cliff_i}$) will be assigned a $p(x \mid \beta, \omega)$ value of 0.5. With this use of $\varphi = 0.5$ as a reduction factor, area iterativly identified as



ice cliff will have a $p(x \mid \beta, \omega)$ a factor of 0.5 greater than any other area but no steep surface slope will be completely rejected as having $p(x \mid \beta, \omega) = 0$.

The result of this iterative process is $90/n$ gridded ice cliff probability maps and vector ice cliff shapes ($A_{cliff_i}$) for the entire spatial domain.

## 3.2.2 Heuristic selection of $\beta_{opt}$: a best $\beta_i$

The $90/n$ ice cliff probability maps ($\beta_i$) are constrained by two conditions: where the spatial domain is unrealistically dense with ice cliff and where there is zero ice cliff area. Using the vector ice cliff shape area ($A_{cliff_i}$), ice cliff fraction,$y_i$, is calculated as $A_{cliff_i}/area(spatial\,domain)$. To derive a continuous, functional form of ice cliff fraction, a single-tailed ($\beta \nless 0$) Gaussian distribution function,

$$y(\beta) = a \exp\left( - \left(\frac{\beta - b}{c}\right)^2 \right),\qquad(2)$$

is fit to $y_i$ and $\beta_i$ where $a$, $b$ and $c$ are fitting parameters. The curve expresses unrealistically high $y(\beta)$ with low values of $\beta$ and, if ice cliffs do exist within the spatial domain, the slope of $y(\beta)$, $y'(\beta)$, approaches 0 as $\beta$ approaches 90°. If there are no ice cliffs within the spatial domain, the curve will reach $y(\beta) = 0$ without a long tail (ending further iterations). This truncation is likely the key distinction of areas with no ice cliffs relative to ice cliff abundant domains (see Sect. 5.2). If there are ice cliffs within the spatial domain, some value of $\beta_i$ will best match the true ice cliff fraction. The method uses a heuristic approach to select this $\beta_i$, termed $\beta_{opt}$, which might provide the most accurate final $A_{cliff}$ and coupled $p(x \mid \beta, \omega)$ map.

Where $\beta$ is low and $y(\beta)$ is unrealistically high, all ice cliffs will likely be included (high *true positive rate* [defined in Sect. 3.5]) yet will be accompanied by a large amount of non-ice cliff area (low *precision* [defined in Sect. 3.5]). Conversely, as $\beta$ approaches 90°, or $max(\beta)$ if $max(\beta) < 90°$, the small areas within $y(\beta)$ will very likely be ice cliff area (high *precision*), but widely under resolve the true ice cliff area. In the absence of validation data to explicitly optimize *true positive rate* and *precision*, the 'mid' point or 'elbow' of the curve as $y(\beta)$ shifts from a steep slope, high $y'(\beta)$, to $y'(\beta)$ approaching 0, is hypothesized to correspond to the optimized maximum of both *true positive rate* and *precision*. This point, $(\beta_{opt}, y_{opt})$ is defined as

$$(\beta_{opt}, y_{opt}) = (\beta, y(\beta))\ where\ (\beta, y(\beta)) \cap max(d),\qquad(3)$$

where

$$d = distance(P_1, P_2, (\beta, y(\beta))) = \frac{\mid (y_2 - y_1)\beta_i - (\beta_2 - \beta_1)y + \beta_2 y_1 - y_2 \beta_1 \mid}{\sqrt{(y_2 - y_1)^2 + (\beta_2 - \beta_1)^2}}.\qquad(4)$$





$d$ is the orthogonal distance from a line defined by points $P_1$ and $P_2$ to the function $y(\beta)$ (Eq. (2)), where

$$P_1 = (\beta_1, y_1)$$
$$\beta_1 = \beta \ where \ y'(\beta) = \gamma$$
$$y_1 = y(\beta_1)$$

$$(5)$$

and

$$P_2 = (\beta_2, y_2)$$
$$\beta_2 = \beta \ where \ y(\beta) = \ max(y(\beta))$$
$$y_2 = y(\beta_2).$$

$$(6)$$

$\gamma$ is an input parameter defining the limit when $y'(\beta)$ is effectively 0. Since the function asymptotically approaches 0, without $\gamma$, $\beta_1$ would always be 90° and this geometric approach would likely fail to identify the so called 'mid' point or 'elbow' of the curve. Additionally, DEM errors can sometimes have vertical or near vertical slopes (e.g. a raised artifact). These errors will be identified as ice cliff area but will not impede the calculation of $(\beta_{opt}, y_{opt})$ because very steep area causes a vertical translation of the function $y(\beta)$, thus not affecting $y'(\beta)$.

Finally, the iterative method is again run for one more iteration where $\beta_i = \beta_{opt}$ to produce a final automated cliff probability map. If visual inspection suggests large errors, all of the cliff probability maps and vector ice cliff shapes generated from the earlier set of iterations are retained and can be assessed to establish if a more adequate $\beta_i$ value should be considered optimal.

### 3.3   Domain segmentation for large areas

Using this method over large spatial domains might be computationally demanding on typical desktop or laptop computers. To
address this, a precursory function segments large domains into less computationally taxing tiles. Because the ice cliff mapping depends on statistics calculated across the entire area considered, it is critical that no segment be small so that statistics cannot be meaningfully computed. A target/maximum spatial domain is defined by the user as the length of an edge of a square, $L_t$. If the debris covered area of interest is below $L_t{}^2$, no segmentation will be applied. If the debris covered area is greater than $L_t{}^2$, the debris covered area is subdivided by a square grid with side length $L_t$. The function finds the area of debris cover
occupying each grid cell and attempts to merge neighboring cells one at a time until their fractional debris-covered area sum to 1. A look distance factor in number of $L_t \times L_t$ cells, $n_c$, controls if and how far the tool will look beyond empty space or cells where an unsuccessful match (summed fraction > 1) occurred and still be considered as 'neighboring' cells. When a cell or set of previously merged cells have (1) exhausted the set of possible neighboring cells within the look distance, and (2) all





have returned a summed debris cover fraction greater than 1, the cell or set of cells are defined as a closed tile that will become an input spatial domain to the iterative and heuristic optimization scheme. The tool automatically identifies and individually processes each tile and a final merged product of both gridded $p(x \mid \beta, \omega)$ and a vector shapefile corresponding $A_{cliff}$ are generated.

## 3.4 Derivation of a calibration dataset for Canwell Glacier

To calibrate a method that automatically maps ice cliff area, a sufficiently accurate 'truth' dataset, $x$, is needed. We define $x$ as the area defined manually by digitizing ice cliff outlines from the high resolution visible and thermal data described in Sect. 2.2.1. Elevation data described in Sect. 2.2.1 were not explicitly used to digitize from but were used in a 3D viewer with draped visible and thermal layers to assess generated ice cliff outline quality. Area that was clearly ice cliff in visible and thermal data but not apparent in elevation data (possibly due to errors in the DEM) was still mapped as ice cliff area. Given the ambiguities described in Sect. 1.1 regarding "thin" debris cover, ice cliffs were liberally outlined, including, for example, cliffs that where nearly 100% covered by debris, yet had a unique thermal signature relative to the surrounding debris cover indicating thinner debris. No minimum size was considered, thus ice cliffs below the resolution of the tool input data are penalized in quality assessment metrics, if missed.

True ice cliff surface area (as opposed to map view surface area) was calculated by summing the secant of slope pixels derived from DEM data within $x$ and multiplying by the area of a DEM pixel.

## 3.5 Statistical measures of performance

A suite of statistical measures of performance are needed to isolate and illustrate performance trade-offs and rank success between parameter sets and alternative, less complex methods. All performance metrics are calculated using the tool output vector shape defined by $A_{cliff}$ rather than distributed $p(x \mid \beta, \omega)$. True positive ($TP$) is defined as area where the ice cliff mapping technique output intersect true ice cliff area, $x$. True negative ($TN$) is defined as area where the method did not identify ice cliffs that intersects true non-ice cliff area (debris-cover area), $x^c$. False positive ($FP$) and false negative ($FN$) are defined as area identified as ice cliff that is not included in $x$ and ice cliff area present in $x$ but absent in the automated output, respectively. Using these quantities, the following common metrics (e.g. Fawcett, 2006) are defined:

$$true\ positive\ rate = \frac{TP}{TP + FN} \tag{7}$$

$$precision = \frac{TP}{TP + FP} \tag{8}$$

$$accuracy = \frac{TP + TN}{TP + FP + FN + TN}. \tag{9}$$





*True positive rate* (also called recall) is the ratio of successful classification over total true ice cliff area and *precision* measures the probability that area identified as ice cliff is in fact ice cliff. Ideally, these metrics should be equal to each other and, if the cliff mapping is perfect, both equal 1. *Accuracy* is the proportion of true results, but because ice cliffs will most often occupy a small fraction of a debris-covered area and *accuracy* accounts for $TN$ as well as $TP$, it becomes a less informative metric. For example, if 1% of a debris-covered portion of a glacier is ice cliff, not mapping ice cliffs at all will yield an ice cliff mapping *accuracy* of 99%. We therefore introduce two additional metrics, outside of the standard suite of statistical measures of performance, that are independent of $TN$ and thus help evaluate ice cliff mapping success:

$$error\ distribution = \frac{FP}{FN} \tag{10}$$

$$error\ magnitude = \frac{FP + FN}{TP + FN}. \tag{11}$$

*Error distribution* provides a measure of balance between $FP$ and $FN$ errors. An *error distribution* $> 1$ means there is more debris-covered area mapped erroneously as ice cliff than ice cliff area erroneously mapped as debris-covered and vice versa for *error distribution* $< 1$. Ideally, *error distribution* is 1. *Error magnitude* is a ratio of the total erroneously mapped area, both $FP$ and $FN$, over the true ice cliff area ($x$, which is equal to the sum of $TP$ and $FP$). If *Error magnitude* is 0 ice cliffs are perfectly mapped, if *error magnitude* is, for example, 2, then error is a factor of 2 greater in spatial extent than the true ice cliff area.

## 3.6 Calibration

The method presented here requires five key input parameters to map ice cliffs: $L_e$, $\alpha$, $\beta_e$, $A_{min}$ and $\gamma$ (Table 1, described in Sect. 3.2.1, bold font quantities in Fig. 4). A matrix of different parameter sets were tested using Canwell Glacier data. The success of each parameter set was quantified against the manually generated ice cliff outlines described in Sect. 3.4 using the statistical measures of performance derived in Sect. 3.5.

## 3.7 Quantification of "thin" debris cover on ice cliffs

As described in Sect. 1.1, ice cliffs can be considerably covered by rock fragments. The automated method presented here uses elevation data alone and therefore depends on the assumption that steep terrain within a debris cover is ice cliff. Because the validation data described in Sect. 3.4 is in part visible data, assessment of "thin" debris cover on ice cliffs can be made and used to further interpret automated ice cliff mapping results. Here, the word "thin" is used colloquially under the premise that an ice surface able to retain large clasts and/or a debris cover equal or greater in thickness to the surrounding debris-covered area is not an ice cliff, leaving dust to small clasts to constitute "thin" debris covering on ice cliff faces. The optical orthomosaic was converted to greyscale and a pixel value threshold was selected manually that discriminated between debris-free and debris-covered area. This process is similar to the satellite data based method described in Sect. 3.1 to identify debris-covered area,



but without a band ratio correction and at a much higher spatial resolution of ~8 cm. The results provide the distribution and fraction of debris cover on ice cliffs (Fig. 7).

# 4 Results

## 4.1 Canwell Glacier

The manually generated, 'true' ice cliff dataset, $x$, shows 4.9% of the 1.7 km$^2$ Canwell Glacier study area is ice cliff in map view (not considering slope). Ice cliff map view area (84,630 m$^2$) under represents true ice cliff surface area (104,920 m$^2$, considering slope) by 19%. Table 2 shows the parameter set matrix described in Sect. 3.6. The results summarized in Table 2 (1) suggest that the method is stable and robust because no parameter set produced exceptionally poor results (e.g. computed *error magnutude* across all parameters sets fall within a range of 0.94 to 1.06); and (2) allow the selection of a

'best' parameter set, selected with an emphasis on high and equal values of *true positive rate* and *precision*: $L_e$ = 10 m, $\alpha$ = 3.54 m, $\beta_e$ = 3° and $A_{min}$ = 250 m$^2$. $\gamma$ is excluded from Table 2 because after testing different values of $\gamma$, a value of 0.0001 consistently returned results where *true positive rate* and *precision* are close to equal. Calibration results in Table 2 are shown alongside the same statistical measures of performance applied to a range of simple surface slope thresholds used to identify ice cliffs. This comparison shows that a carefully selected slope threshold can provide an ice cliff map comparable

in statistical measure to the more complex method presented here. However, simple slope threshold results are shown to be sensitive with respect to the slope value used and would require additional data, similar in quality to those described in Sect. 3.4, to validate the threshold selection and subsequent results.

    Using the parameter set [$L_e$ = 10 m, $\alpha$ = 3.54 m, $\beta_e$ = 3°, $A_{min}$ = 250 m$^2$ and $\gamma$ = 0.0001] as input, Fig. 6 shows the heuristic approach for selecting $\beta_{opt}$, a best $\beta_i$. While the location of ($\beta_{opt}, y_{opt}$) is dependent on $\gamma$ and therefore not mathematically

robust, the close coincidence of $\beta_{opt}$ and $\beta$ where *true positive rate* and *precision* intersect (Fig. 6) suggests the technique performs as intended. *Error magnitude* deviated only slightly from a value of 1 in all of the parameter set tests. This indicates the best results of this method are incurring errors equal in area to the true area of ice cliffs. This is a non-trivial error but likely an unavoidable trade off for a method designed for wide scales and modest data input. It is important to note that comparing only percent ice cliff area values between measured, $x$ (4.9%), and modeled (5.3%) would give a misleading perception of very

low error, 0.4%, when the true error is closer to 5% ($1 - accuracy$) if considering the entire domain or higher if considering only ice cliff area error (*true positive rate* and *precision*). For most tests, errors are fairly distributed between FP and FNs and shown by *error distribution* having a proximity to 1. As described in Sect. 3.5, *accuracy* is a poor indicator of success mapping a feature that occupies a small fraction of a total area, favoring a setting of higher *precision* over *true positive rate*, but it is the only metric used in this study that rewards $TN$ area and quantifies overall performance.

Results using the best parameter set for Canwell Glacier have an *accuracy* of 0.952, where $FP$ and $FN$ errors are close to evenly distributed with an *error distribution* = 1.14 and an *error magnitude* = 0.98, which is slightly below the mean for all tested parameter sets (0.99). *True positive rate* and *accuracy* are higher (0.54 and 0.51, respectively) than those achieved by the best simple slope threshold, 27° (0.49 and 0.50, respectively). These results suggest that the method presented here





can achieve results that are slightly more accurate than the best simple slope threshold and is far more robust: sensitivity to changing parameters is low while different simple slope threshold values produce wider variance in statistical measures of performance (Table 2) and are thus more sensitive.

### 4.1.1 Mapping success in the context of ice cliff characteristics

Figure 7 shows mean surface slope, map view surface area and the percentage of "thin" debris covering every ice cliff mapped in $x$. These characteristics are shown in the context of *true positive rate* (Eq. (7)) now calculated for every ice cliff (*true positive rate* mentioned in all instances prior was calculated for all ice cliffs together), and $FP$ ice cliffs, which we define as isolated shapes that are solely $FP$ area and do not share a boundary with a shape in $x$. The figure shows that, in this portion of Canwell Glacier, most of the "cleanest" ice cliffs are still covered by a non-trivial (>50%) amount of debris. The

percentage of "thin" debris cover on ice cliff faces appears continuous above 50%, such that there is no clear boundary in the data that could define what is and is not an ice cliff if the percentage of "thin" debris cover was the main deterministic variable.

Figure 7 provides a wider context to what is present in reality and what is detected as an ice cliff by the automated method presented here. The figure shows the limitations of detecting ice cliffs where steep surface slopes are either not present or not resolved in the data.

## 4.2 Ngozumpa Glacier

Table 3 shows the performance of the best parameter set found for Canwell Glacier applied to the lower ablation zone of Ngozumpa Glacier. There is a decrease in performance, for example, *true positive rate*, *precision* and *error magnitude* for Canwell Glacier are 0.54, 0.51 and 0.98, respectively, and 0.53 ($-1$%), 0.32 ($-19$%) and 1.58 ($+0.6$), respectively, for Ngozumpa Glacier. The imbalance between *true positive rate* and *precision* indicates the automatically selected $\beta_{opt}$ does 

not coincide with the ideal $\beta_i$ where *true positive rate* and *precision* are optimized. However, in the context of Table 3, $\beta_{opt}$ was off by only $\pm2.5°$ (*true positive rate* and *precision* constrained between simple surface slope thresholds $35°$ and $40°$). Considering that not a single alteration was made to the model input parameters, the method mitigated the different surface type (very hummocky surface relative to Canwell Glacier, see Fig. 3) and different DEM generation method (satellite based rather than airborne structure from motion) far better than a simple slope threshold found at Canwell Glacier and assumed to

be transferable to Ngozumpa Glacier (Tables 2 and 3). The debris covered area considered on Ngozumpa Glacier was broken into three processing tiles where $\beta_{opt}$ found for each tile was $30.8°$, $32.5°$, and $33.5°$, while $\beta_{opt}$ for Canwell Glacier was $23.5°$. The ~$10°$ difference between $\beta_{opt}$ for both glaciers is significant but appropriate for each location as shown in Fig. 3 or observed by comparing the statistical measures of performance for the range of simple slope threshold values shown in the far right column of Tables 2 and 3. Taking the best simple slope threshold of $27°$ found for Canwell Glacier and applying it

to Ngozumpa Glacier would result in an ice cliff map with a *precision* of 0.10 ($-41$% relative to Canwell Glacier results) and area mapped incorrectly a factor of 7.91 ($+6.93$) times greater than the true ice cliff surface area (Table 3). Using the method presented here and the parameters from Canwell Glacier, *true positive rate* and *precision* are closer to balanced and





erroneously mapped area is only a factor of 1.58 greater than the true ice cliff area, comparable in magnitude (+0.6) to the best results for Canwell Glacier.

The results shown in Table 3 were generated without removing area identified in Thompson et al. (2016) as having poorly resolved surface elevation. Poorly resolved on-glacier area was predominantly at locations that were shaded by cast shadows and thus often concurrent with steep terrain and ice cliffs. Visual inspection of the data suggested that while the computed

precision might be low, the well resolved area above and below steep terrain combined with the absence of extreme outliers resulted in steep terrain still being resolved as steep. Acknowledging the uncertainties within this context, we also testing the method with poorly resolved elevation locations removed from both the DEM and the manually generated 'true' ice cliff area. These results produced a reduction in *true positive rate* (0.44) yet more in balance with *precision* (0.38) and a comparable *error magnitude* (1.62).

## 5 Discussion

This method attempts to resolve two key components in DEM-based automated ice cliff mapping that are apparent in the histogram in Fig. 3: (1) selecting a threshold that discriminates between ice cliff and debris covered area; and (2) adding/removing the correct area that is within the surface slope overlap between ice cliff and debris-covered areas. The method presented here attempts to add low slope ice cliff area by looking beyond the ends of the ice cliffs and rejects area with the same (low) slope

that is not neighboring the ends of an ice cliff. Rejecting area that is steep but not truly ice cliff is difficult using elevation data alone. The only mechanism to remove this area within this method is eliminating mapped ice cliff area below a threshold, however this is a delicate balance between reducing errors and reducing resolution to which the tool can resolve ice cliffs. The abundance of small, low angle ice cliffs with a very low *true positive rate* shown in Fig. 7 indicate improvements to automated ice cliff detection will need to, in part, focus on small ice cliffs whose surface slope may be saturated due to data

resolution.

### 5.1 Alternative approach

Alternative methods were tested before selecting the presented method as best. One alternative approach used optical satellite data to accept or reject potential ice cliff area. A main objective in this approach was identifying and removing the $FP$ ice cliffs clustered at the top of Fig. 7. The Landsat 15 m panchromatic band was corrected for illumination variance from

25 topography using the Minnaret method (Smith et al., 1980). Bright and dark regions of the image were then identified as area $mean(L_H) \pm std(L_H)m$ where $L_H$ is Minnaret corrected radiance values and $m$ is a model parameter. These regions were established with the assumption that ice cliffs with an aspect that is illuminated by the sun would be optically bright relative to surrounding debris cover and ice cliffs with an aspect that is in cast shadow would be optically dark relative to surrounding debris cover. A buffer sequence was then applied to separate shapes that were narrowly attached (e.g. hourglass shaped): shapes

were uniformly shrunk and expanded by a user defined distance. A shape that contained $area > 0$ of both seed slope area (e.g. $area > \beta_u$ [$\beta_u$ is defined in Sect. 3.2.1]) and optically bright or dark area would be assigned a high ice cliff likelihood.





However, this method failed to perform better than a method using surface slopes alone because of two linked factors: data resolution and "thin" debris covering true ice cliffs. Figure 7 shows that even the "cleanest" ice cliffs are still around 50% debris covered. A 15 m optical pixel perfectly centered on an ice cliff with 50% debris cover should have a distinguishable signal, but this is a best case in both pixel/ice cliff location coincidence and fraction of "thin" debris cover. It is clear to see how using 15 m resolution data will quickly fail to resolve smaller and more debris-covered ice cliffs. The buffer sequence to

separate narrowly attached shapes had the positive (intended) result of detaching narrowly joined debris-covered area and ice cliff area, allowing ice cliff area to be considered separately and positively identified as an ice cliff while rejecting the debris covered area. However, the shrinking step had the negative (and more frequent) effect of completely removing shapes that where, in at least one dimension, equal to or less than $2\times$ the user defined buffer distance.

## 5.2   Wider application

The testing of this method at two locations on opposite sides of the Earth with the same input parameters suggests the method can be applied/transferred elsewhere with little loss of performance. Figure 8 shows repeat runs for the Canwell Glacier with all parameters constant only resampling the DEM to different resolutions. This offers a first order estimate of how performance will decline with coarsening DEM data resolution; however, it is possible that a recalibration of the input parameters could improve results when using data at lower resolutions.

To examine how this method performs for a debris-covered area with no ice cliffs, $TP$ and $FP$ area was removed from the Canwell Glacier spatial domain. Because $FP$ area is already identified as area where the tool fails, we removed this area also so that all remaining area is ice cliff free with maximum confidence. Figure 9 shows the resulting plot for selecting $\beta_{opt}$. The method still (erroneously) identified ice cliffs; however, the shape of $y(\beta)$ abruptly terminates rather than slowly approaching zero. This is the key characteristic that would suggest there are actually no ice cliffs within the spatial domain. A domain

with ice cliffs will likely have at least a few very steep areas that will carry computations through higher values of $\beta_i$, while simple undulating surface topography should not produce abruptly steep slope values and as soon as $\beta_i$ exceeds the maximum slope present, iterations will stop (Fig. 9). Due to the abrupt termination of the curve derived where no ice cliffs are present in reality, a high measure of linear correlation, e.g. Pearson correlation coefficient ($r$), between $\beta_i$ and $y_i$ might offer an automated binary classification of whether a spatial domain does or does not contain ice cliffs. Further testing would be required to derive

a threshold value that could be confidently applied to other data, but the data shown in Fig. 9 supports this hypothesis where $r = -0.98$ for a spatial domain with no ice cliffs and $r = -0.91$ for a spatial domain with ice cliffs.

## 5.3   What are ice cliffs and how should glaciologists map them?

Throughout this paper the ambiguity of what is an ice cliff has been mentioned. Figure 10 shows an example of an area possibly in the process of becoming an ice cliff. The relief of this medial moraine has facilitated a setting where mass wasting exceeded

englacial debris exhumation and accumulation. The area remains evenly and largely debris-covered, but does appear distinct both in visible and thermal imagery. However, if this is considered sufficient criteria to be defined as an ice cliff, there is no clear edge where ice cliff ends and debris cover begins. We had a difficult time deciding if this feature should be included in

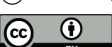



the 'truth' dataset ($x$), ultimately deciding to include it (and similarly ambiguous features), but also with the caveat in Sect. 3.4 stating that "ice cliffs were liberally outlined". The slope of the feature identified in Fig. 10 was sufficient to be identified by the automated method as an ice cliff while the surrounding slopes of the medial moraine were not.

Figure 11 shows a second example where an ice cliff could be mapped in detail excluding bands of debris within the ice cliff face (Fig. 11d) or could be mapped more broadly including the debris bands (Fig. 11c). This is a setting where time could

be considered when defining an ice cliff as described in Sect. 1.1. If these debris bands are long term fixtures and there are sufficient data to resolve the individual ice cliff faces then possibly the detailed mapping is correct. However, if the clasts within the debris bands are transported to the ice cliff margins and not resupplied the more broad/coarse mapping approach would be appropriate. For this study, we delineated ice cliff(s) 1 from Fig. 11 using the more coarse mapping approach.

While higher resolution data capturing a suite of properties (e.g. visible and surface temperature data) can further resolve

what is truly present, it will likely not resolve classification ambiguities (e.g. Fig. 10 and 11), and at the present time these data are not available at large scales. Focus should therefore be more targeted towards mapping consistency. This is best met when automated methods can be applied with sufficient levels of confidence, eliminating technician bias and error.

For both automated and manual mapping methods used this study we map ice cliffs as a 2D area or a 3D surface. An alternative method to map ice cliffs is to define a line along the ice cliff top edge (e.g. Thompson et al., 2016; Watson et al.,

2017). Figure 11 provides a comparison of all three methods (3D surface area, 2D map view surface area and 1D top edge line) and shows that when mapping ice cliffs as a 2- or 3D area, refined detail leads to a refined (smaller/more accurate) area, while refined detail leads to an expansion of ice cliff top edge length when mapping ice cliffs in 1D. This suggests that 2- or 3D area is a more reliable measure and likely more communicative if drawing a comparison with other areas or regions or studies.

## 6    Conclusions

This study presents a new automated method for mapping ice cliffs within supraglacial debris cover. The method uses glacier outlines and satellite imagery to isolate debris-covered area where ice cliffs might exist and then uses DEM data alone to map ice cliff area. The DEMs used in this study had a spatial resolution of 5 m. The method is designed to accommodate regional variability in ice cliff characteristics by selecting unique surface slope threshold values automatically. The method also attempts to improve performance by explicitly considering the often narrowing ends of ice cliffs. The method was calibrated using

data from Canwell Glacier in Alaska, USA, and validated using data from Ngozumpa Glacier in Nepal. The best parameter set for Canwell Glacier produced an ice cliff map with essentially equivalent success to a carefully selected simple surface slope threshold, which itself carried a degree of error with *true positive rate* and *precision* both around 0.5. While the application of a simple surface slope threshold is a much easier mapping technique, the selection of a sufficient threshold requires supplemental, high resolution data due to the rapid increase in error with less ideal threshold selections. The method

presented here reduces this instability and offers more confident ice cliff mapping when supplemental data are not available. With no parameter alteration, the method was applied on the other side of the world. Results from Ngozumpa Glacier show a decrease in performance relative to Canwell Glacier where *true positive rate* is similar but *precision* is 19% less. This





is still however, an ice cliff map with more success than if the carefully selected simple surface slope threshold for Canwell Glacier was assumed to be transferable to Ngozumpa glacier. Under this assumption, $precision$ is 41% less and the ice cliff area mapped incorrectly is a factor of 6.93 more than the results for Canwell Glacier. We therefore conclude that simple surface slope thresholds (1) carry a non trivial degree of error even if carefully selected; and (2) cannot be considered to be transferable to other regions. In this study we have quantified (1) and presented a method to mitigate (2).

While we only consider two locations, these results offer an idea of how well the method might perform in other regions without supplemental validation data and opens the possibility for deriving ice cliff area at large scales. With a DEM of adequate spatial resolution, which we show is best if around 5 m, and sufficient computational capacity, this tool could be applied to all glacierized area on Earth. Additionally, running the tool on temporal data will produce a time-lapse evolution of ice cliff formation, cessation, melt patterns and motion through the glacier flow regime. Further validation of this method in

other regions will help to either support or discredit our claim of wide applicability. Ambiguities defining what is an ice cliff are likely to persist in any technique used to map ice cliffs, but map variance from these ambiguities will become less of a factor if a consistent methodology is used.

## 7   Code availability

[pending publication] Code from this study is available within the Python/ArcPy(ArcGIS) ensemble Debris Cover Tools, which

are open source and available at https://github.com/samherreid.

*Acknowledgements.*  This work was supported by a studentship from Northumbria University. The Technology Advisory Board at the University of Alaska Fairbanks provided funding for the FLIR thermal camera and the Canon camera was provided on loan by the University of Alaska Fairbanks Rasmuson Library. We thank Rebekah Tsigonis for her help in the field, Alex Shapiro at Alaska Land Exploration, LLC for saving us from having to bike all of the science gear into the Alaska Range, Sarah Roeske for letting us helicopter hitchhike alongside

her fieldwork, Sarah Thompson for sharing her data and ice cliff map from Ngozumpa Glacier, Ben Brock for support early in the project and Markus Holzner for his clever solution to an optimization problem.



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





**Table 1.** Model parameters. Where $\beta^*_i$ is the surface slope threshold that defines the basic ice cliff shape($A_i$ for each iteration, $A_{ei}$ is the same as $A_i$ but with a slightly lower surface slope threshold, $p(x \mid \beta, \omega)$ is ice cliff probability given surface slope and overlap with $A_{cliff}$ (Sect. 3.2.1), and $y(\beta)$ is the fraction of ice cliffs within the spatial domain as a function of surface slope. Look distance is the number of $L_t \times L_t$ cells the routine will look beyond and still consider as 'neighboring' during segmentation of a spatial domain greater than $L_t{}^2$.

| Symbol | Description | Value used in this study |
|---|---|---|
|  | NIR/SWIR threshold for debris mapping (for this study: Landsat 8 OLI5/OLI6) | 1.2 |
|  | Threshold area for bare ice area reclassified as debris-covered area | 2700 m$^2$ |
| $n$ | Number of iterations | 36 (2.5° increments over 90°) |
| $L_e$ | Ice cliff centerline extension length | see Table 2 |
| $\alpha$ | Centerline buffer distance | see Table 2 |
| $\beta_e$ | Degrees by which $\beta^*_i$ is reduced to define $A_{ei}$ | see Table 2 |
| $A_{min}$ | Minimum cliff area threshold | see Table 2 |
| $\varphi$ | $p(x \mid \beta, \omega)$ reduction factor | 0.5 |
| $\gamma$ | Limit where $y'(\beta)$ (Eq. (2)) is effectively 0 | 0.0001 |
| $L_t$ | Target/maximum domain processing square tile side length | 1500 m (area: 2.25 km$^2$) |
| $n_c$ | Look distance for domain segmentation | 1 |





**Table 2.** Model parameter calibration at Canwell Glacier using statistical measures of performance derived in Sect. 3.5. The set of two boxes on the left describe what values are presented throughout the table and their ideal values. $True\ positive\ rate$ is abbreviated here as $TP\ rate$. A well performing parameter set will have values of $true\ positive\ rate$ and $precision$ that are as close to 1 as possible and balanced. The middle set of boxes show performance for varying parameter sets. DEM resolution for this study was 5 m, thus variation of $A_{min}$ was tested at 0, 5 and 10 pixels; $L_e$ at 2× and 4× pixel length; and $\alpha$ at (pixel length $* \sqrt{2}$)/2 and pixel length $* \sqrt{2}$. The set of boxes on the right show performance of simple surface slope threshold mapping of ice cliffs. Values in bold font are the highest ranking parameter sets for both the method described in this paper and the simple surface slope threshold method. Values in blue and red are the best and worst values, respectively, for all boxes in the table.

Legend box:

| TP rate | Precision |
|---|---|
| Accuracy | |
| Error distribution | |
| Error magnitude | |
| 1.00 | 1.00 |
| 1.000 | |
| - | 1.00 |
| 0.00 | |

Middle table:

| | $A_{min} = 0$ m² | $A_{min} = 125$ m² | $A_{min} = 250$ m² |
|---|---|---|---|
| $L_e = 10$ m<br>$\alpha = 3.54$ m<br>$\beta_e = 3°$ | 0.52  0.53<br>0.954<br>0.94<br>0.94 | 0.53  0.52<br>0.954<br>1.02<br>0.95 | 0.51  0.51<br>0.953<br>0.98<br>0.97 |
| $L_e = 20$ m<br>$\alpha = 3.54$ m<br>$\beta_e = 3°$ | 0.52  0.53<br>0.954<br>0.98<br>0.95 | 0.51  0.53<br>0.954<br>0.91<br>0.94 | 0.51  0.50<br>0.951<br>1.06<br>1.01 |
| $L_e = 10$ m<br>$\alpha = 7.07$ m<br>$\beta_e = 3°$ | 0.54  0.50<br>0.952<br>1.14<br>0.99 | 0.53  0.51<br>0.952<br>1.10<br>0.98 | **0.54  0.51**<br>**0.952**<br>**1.14**<br>**0.98** |
| $L_e = 20$ m<br>$\alpha = 7.07$ m<br>$\beta_e = 3°$ | 0.54  0.50<br>0.951<br>1.22<br>1.01 | 0.54  0.50<br>0.951<br>1.16<br>1.00 | 0.54  0.50<br>0.951<br>1.21<br>1.01 |
| $L_e = 10$ m<br>$\alpha = 3.54$ m<br>$\beta_e = 5°$ | 0.53  0.51<br>0.953<br>1.08<br>0.98 | 0.52  0.52<br>0.953<br>1.02<br>0.96 | 0.51  0.53<br>0.954<br>0.93<br>0.94 |
| $L_e = 20$ m<br>$\alpha = 3.54$ m<br>$\beta_e = 5°$ | 0.51  0.51<br>0.952<br>1.01<br>0.98 | 0.53  0.51<br>0.952<br>1.11<br>0.99 | 0.50  0.51<br>0.953<br>0.95<br>0.97 |
| $L_e = 10$ m<br>$\alpha = 7.07$ m<br>$\beta_e = 5°$ | 0.56  0.49<br>0.950<br>1.32<br>1.03 | 0.55  0.49<br>0.950<br>1.29<br>1.02 | 0.54  0.50<br>0.951<br>1.16<br>1.00 |
| $L_e = 20$ m<br>$\alpha = 7.07$ m<br>$\beta_e = 5°$ | 0.56  0.47<br>0.948<br>1.43<br>1.06 | 0.56  0.48<br>0.949<br>1.40<br>1.06 | 0.55  0.48<br>0.950<br>1.29<br>1.03 |

Right table (simple surface slope threshold):

| | TP rate  Precision<br>Accuracy<br>Error distribution<br>Error magnitude |
|---|---|
| 20° | 0.79  0.24<br>0.866<br>12.20<br>2.76 |
| 25° | 0.60  0.41<br>0.938<br>2.22<br>1.28 |
| 27° | **0.49  0.50**<br>**0.952**<br>**0.97**<br>**0.99** |
| 30° | 0.34  0.66<br>0.960<br>0.26<br>0.83 |
| 35° | 0.03  0.75<br>0.956<br>0.04<br>0.91 |
| 40° | 0.03  0.75<br>0.952<br>0.01<br>0.98 |





**Table 3.** Statistical measures of performance for Ngozumpa Glacier. The value arrangement is the same as in Table 2. The lone box on the left is the bold parameter set from Table 2 applied to Ngozumpa Glacier. The set of boxes on the right show performance of simple surface slope threshold mapping of ice cliffs. Values in bold font are the highest ranking parameter sets for both the method described in this paper and the simple surface slope threshold method. Values in blue and red are the best and worst values, respectively, for all boxes in the table. The performance of a simple surface slope threshold at 27° is shown to discredit the possible assumption that the best threshold from Canwell Glacier (Table 2) could have wider applicability to other glaciers.

|  |  |  |  |
|---|---|---|---|
|  |  | 20° | 0.94  0.05 |
|  |  |  | 0.596 |
|  |  |  | 264.34 |
|  |  |  | 16.21 |
|  |  | 25° | 0.89  0.08 |
|  |  |  | 0.749 |
|  |  |  | 88.50 |
|  |  |  | 10.07 |
|  |  | 27° | 0.85  0.10 |
|  |  |  | 0.803 |
|  |  |  | 53.41 |
|  |  |  | 7.91 |
|  | $A_{min} = 250 \text{ m}^2$ | 30° | 0.79  0.14 |
| $L_e = 10 \text{ m}$ | 0.53  0.32 |  | 0.873 |
| $\alpha = 7.07 \text{ m}$ | 0.980 |  | 23.15 |
| $\beta_e = 3°$ | 2.38 |  | 5.12 |
|  | 1.58 | 35° | 0.61  0.27 |
|  |  |  | 0.949 |
|  |  |  | 4.20 |
|  |  |  | 2.03 |
|  |  | 40° | 0.40  0.50 |
|  |  |  | 0.975 |
|  |  |  | 0.67 |
|  |  |  | 1.00 |
|  |  | 45° | 0.26  0.70 |
|  |  |  | 0.979 |
|  |  |  | 0.15 |
|  |  |  | 0.85 |
|  |  | 50° | 0.16  0.77 |
|  |  |  | 0.978 |
|  |  |  | 0.06 |
|  |  |  | 0.89 |





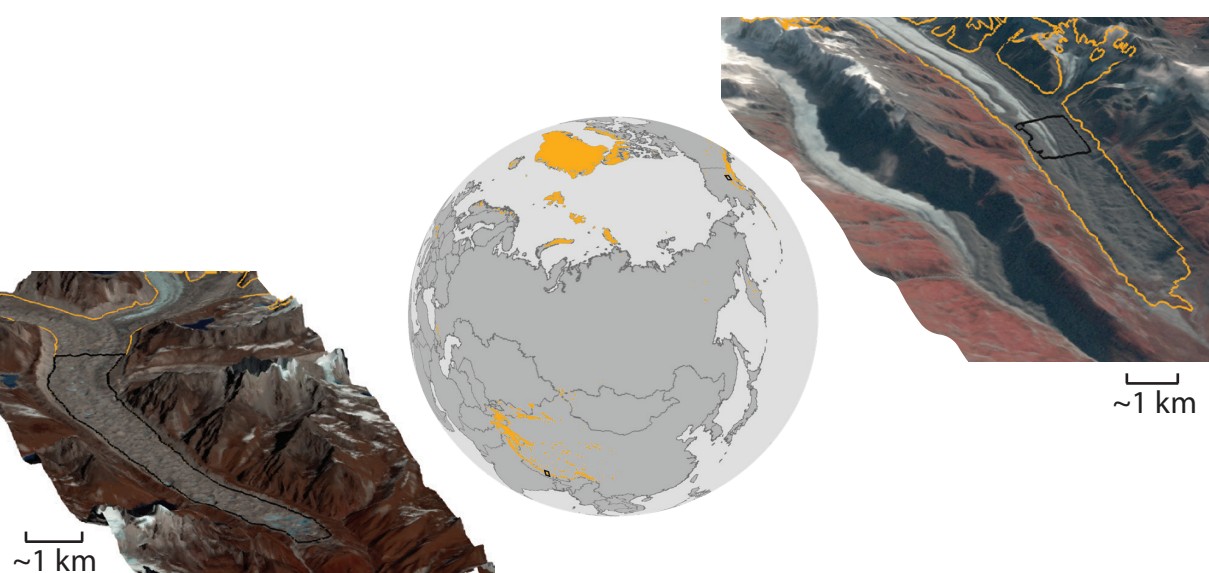

**Figure 1.** Location of Ngozumpa Glacier in the Khumbu Himal, Nepal and Canwell Glacier in the Eastern Alaska Range, Alaska, USA. Global ice cover (Pfeffer et al., 2014; Citterio and Ahlstrøm, 2013) is shown in orange. In the oblique inset maps, orange lines are the glacier extent and the black polygons define the spatial domains used in this study. Ngozumpa Glacier area and base DEM are from Thompson et al. (2016) displayed on a Landsat8 image (path/row: 140/41, acquired on 30 November 2014). Canwell Glacier data are from this study displayed on a Landsat7 image (path/row: 67/16, acquired on the 24 September 2010) draped over a 2010 DEM derived from airborne interferometric synthetic aperture radar (Geographic Information Network of Alaska, GINA; http://ifsar.gina.alaska.edu/).



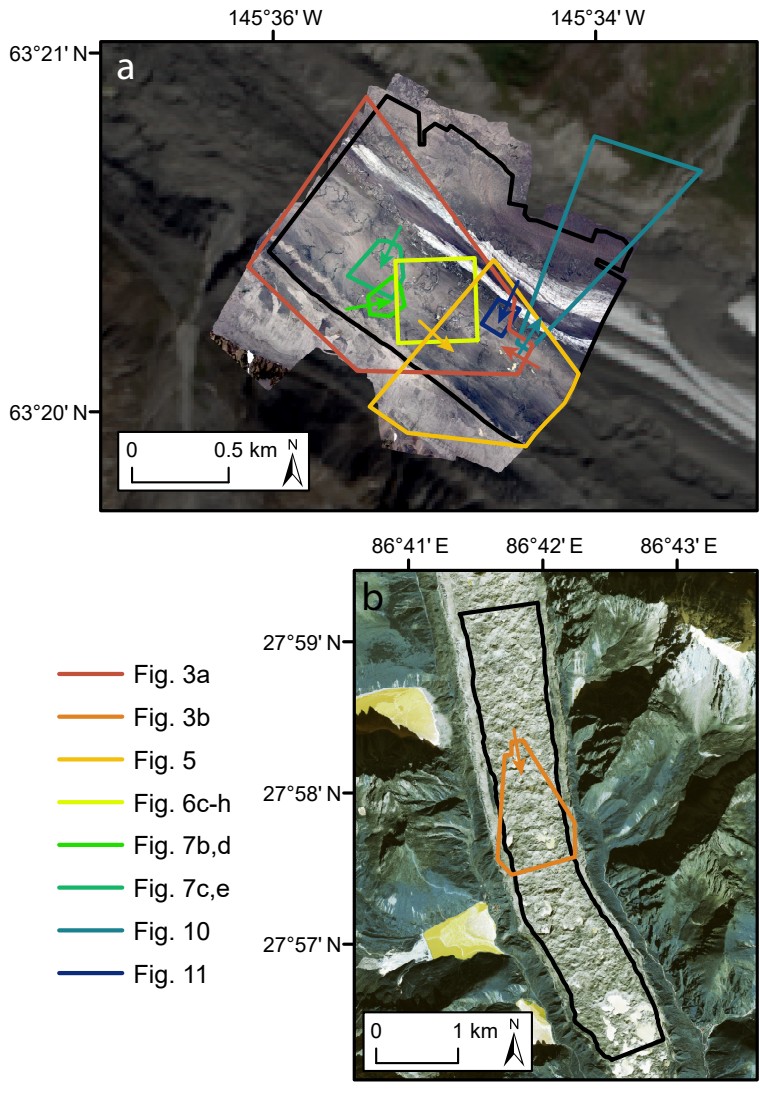

**Figure 2.** Study area location on Canwell Glacier (a) and Ngozumpa Glacier (b). The spatial domains over which the ice cliff mapping method was applied are shown in black. The location of subsequent map-based figures within this paper are shown. Arrows show the look direction of figures that have an oblique orientation. The base image shown in (a) is the orthomosaic collected on 29 July 1016 (Sect. 2.2.1) overlain on a Landsat8 image (path/row: 67/16) acquired on 31 August 2016; and (b) the GeoEye-1 image acquired on 23 December 2012 (Sect. 2.2.2).



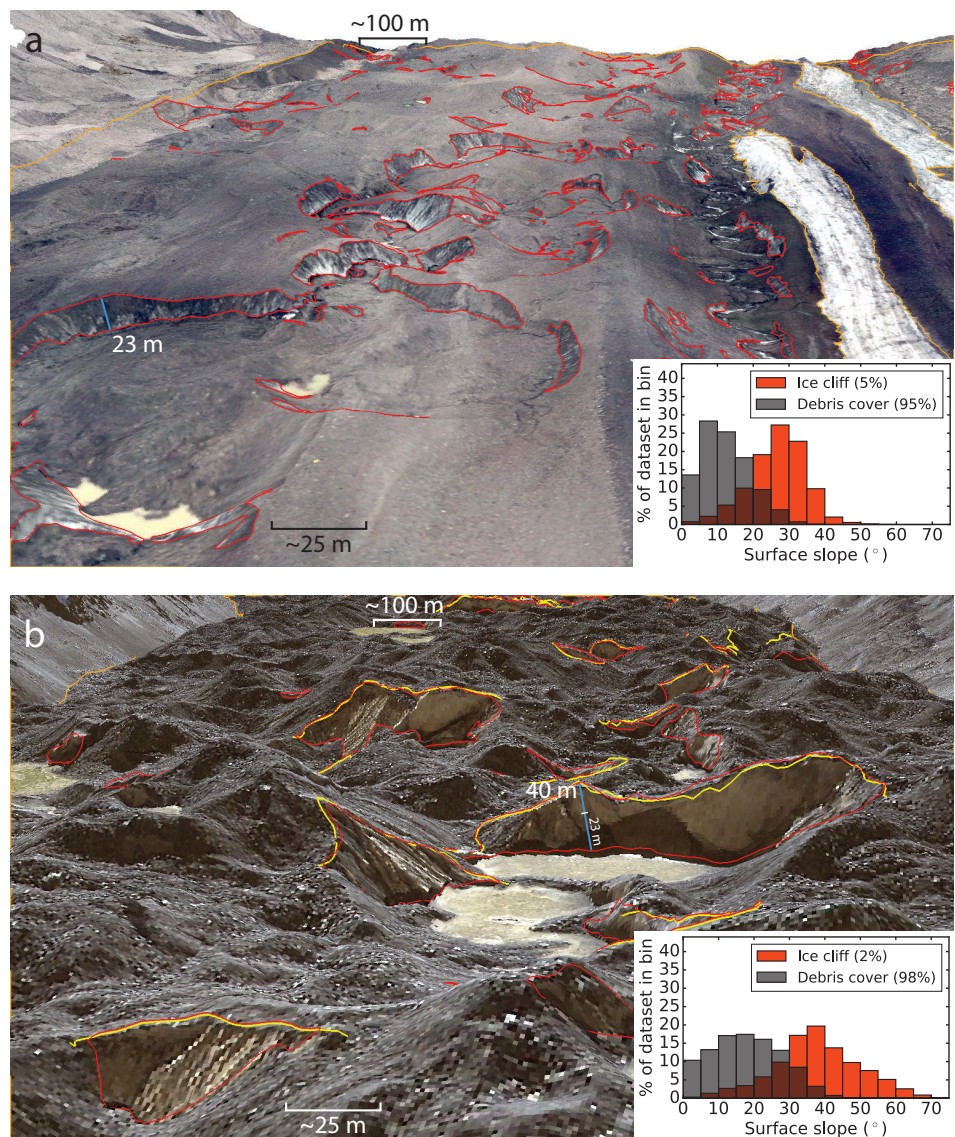

**Figure 3.** (a): portion of Canwell Glacier, Alaska, USA, looking down glacier, a distance of ~1.4 km. Manually generated debris-covered area and ice cliffs ($x$) are outlined in orange and red, respectively. The inset histogram shows the normalized populations of surface slopes present in the pictured debris-covered and ice cliff area. Percentages give the fraction of the total area occupied by both classes (in map view). (b): same as panel (a) but for Ngozumpa Glacier, Nepal, also looking down glacier, a distance of ~1.4 km. The scale in panels (a) and (b) are the same. Yellow lines are ice cliff top edges mapped by Thompson et al. (2016) which were manually expanded to ice cliff area (red) with minor additions. Location shown in Fig. 2.




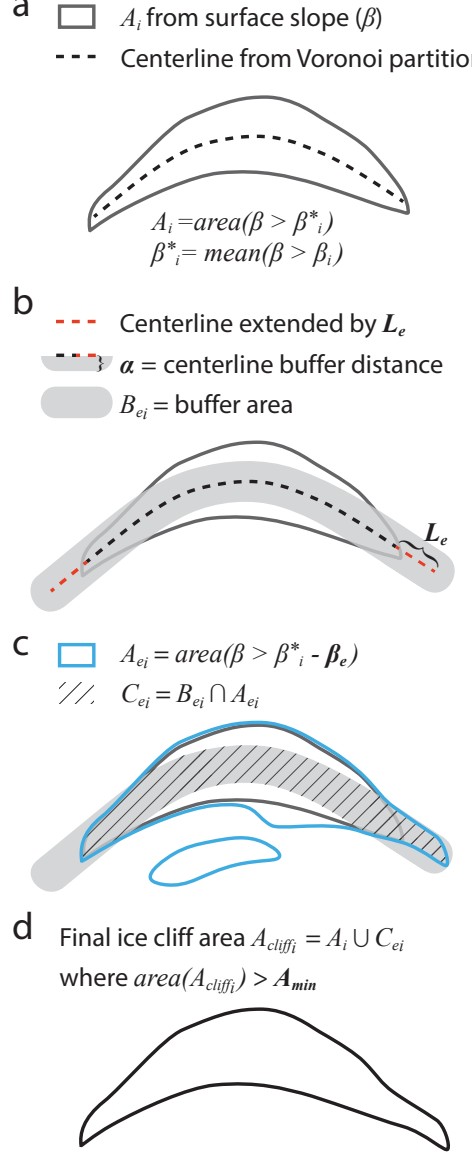

**Figure 4.** Method used to define ice cliff area, $A_{cliff\,i}$ for each iteration, $i$, and a final, optimized iteration, subscript $opt$. Surface slope threshold $\beta_i$ is applied over the range 0-90° from which the subsequent quantities shown in steps a-d are calculated. Quantities in bold font are fixed scalar model parameters that do not vary over the iterative process. (a) shows how the area $A_i$ is defined by $\beta_{*i}$, the mean surface slope of values constrained by $\beta_i$. A procedure using a Voronoi partition defines a centerline within $A_i$. (b) shows this centerline extended by a distance of $L_e$, and transformed into an area, $B_{ei}$, by an outward buffer distance, $\alpha$, applied in all $(x, y)$ directions. (c) shows the definition of $A_{ei}$, an area defined by a surface slope threshold lower than $\beta_{*i}$, where $\beta_{*i}$ is reduced by $\beta_e$. The intersection of $A_{ei}$ and the buffer area, $B_{ei}$, defines $C_{ei}$. $C_{ei}$ allows the identification of ice cliff end area that has a surface slope below $\beta_{*i}$ but above $A_{ei}$ while rejecting area that falls within this same surface slope interval but is not located at the ends of an ice cliff. The intersection of $A_i$ and $C_{ei}$, with areas below a threshold, $A_{min}$, removed defines the final ice cliff area, $A_{cliff\,i}$, for that $i$ (d).



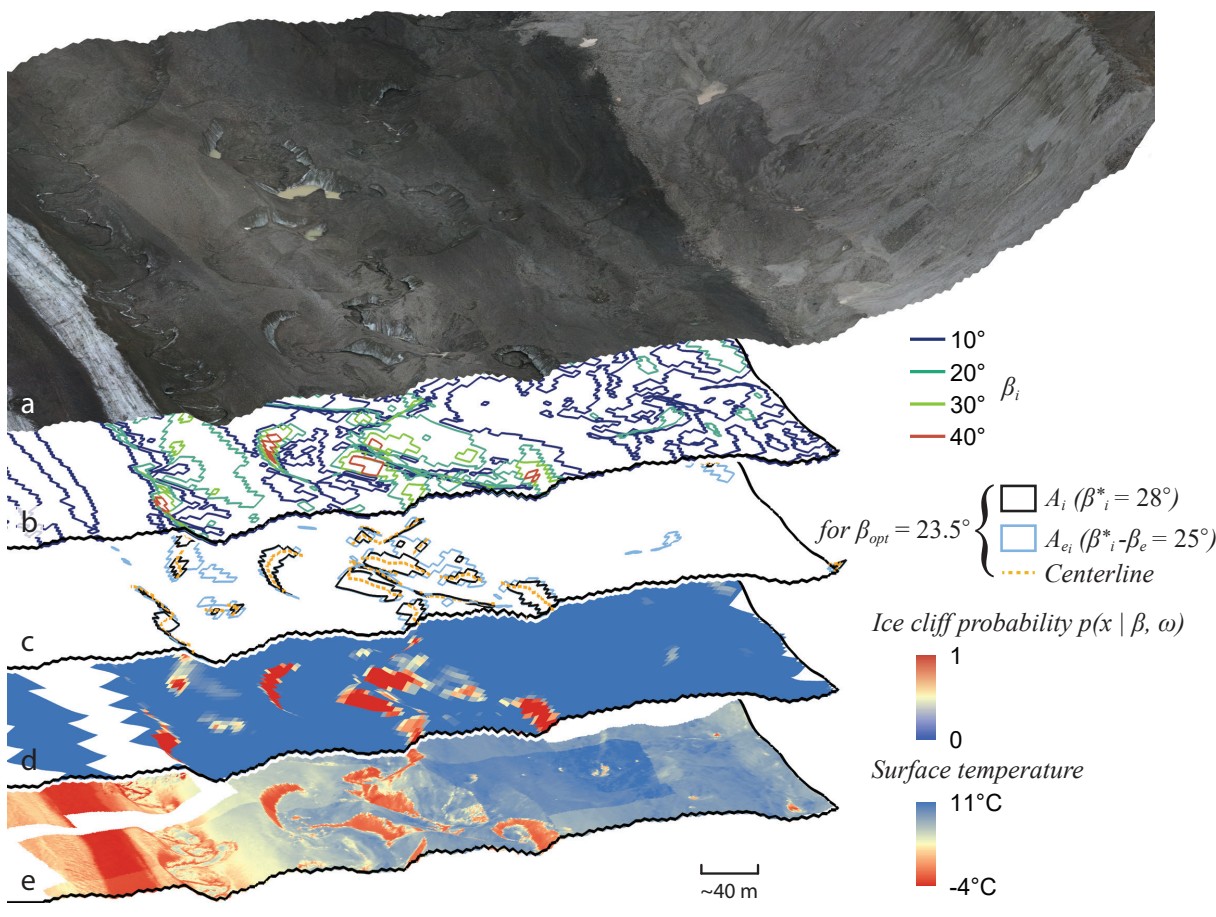

**Figure 5.** (a) orthomosaic of visible imagery collected above Canwell Glacier on 29 July 2016, draped over a DEM derived from the same images. (b) shows the area enclosed for select surface slope thresholds, $\beta_i$, during the iterative process. (c) shows intermediate quantities calculated during the final iteration using $\beta_{opt}$, the optimized $\beta_i$. The area enclosed by $A_{ei}$ and the ice cliff centerlines are used to add low angle area at the ends of ice cliffs to the area $A_i$, the main ice cliff shape defined from $\beta*_i$. (d) shows the final distributed map of $p(x \mid \beta, \omega)$, the computed probability that a given pixel will fall within true ice cliff area, $x$, assigned as a function of surface slope, $\beta$, and $\omega$, the overlap with the final vector ice cliff shape, $A_{cliff}$, generated from the quantities shown in (c). Location shown in Fig. 2.



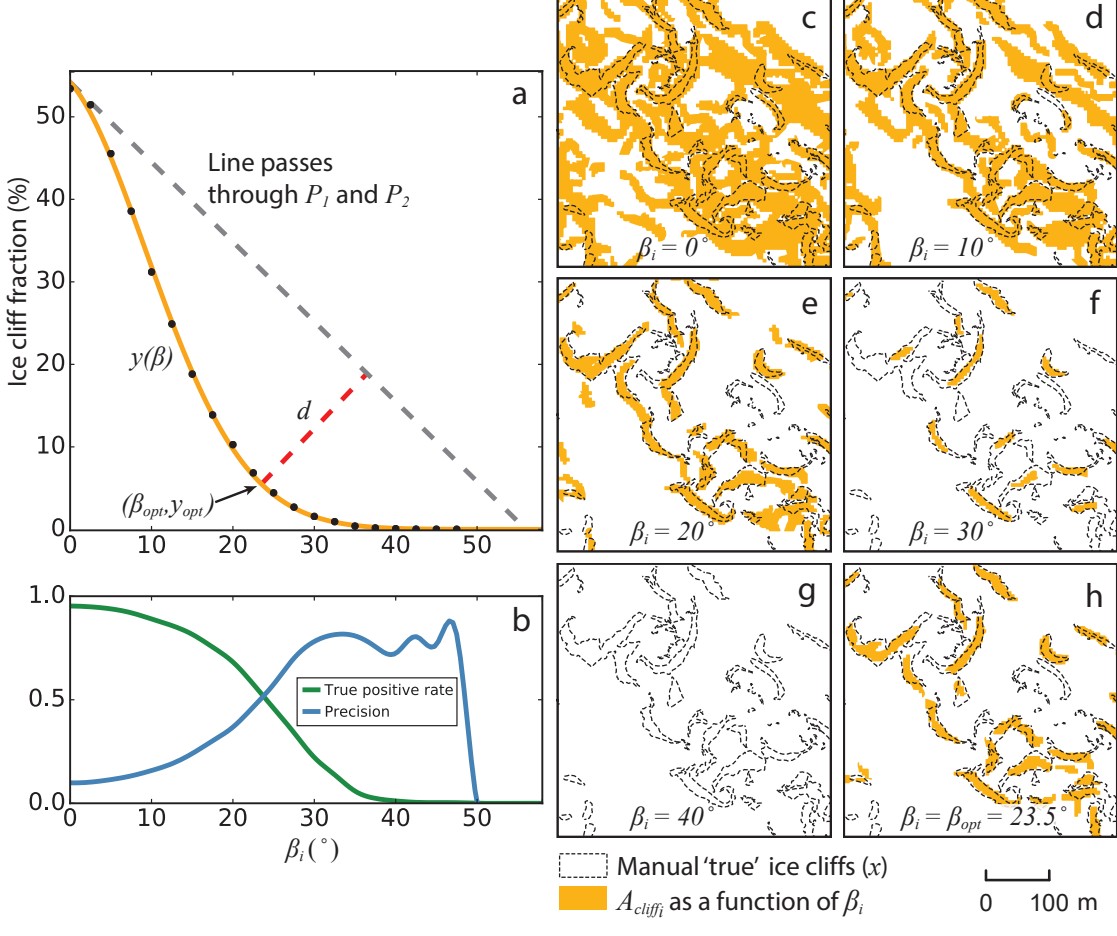

**Figure 6.** (a) shows the heuristic method for selecting $\beta_{opt}$. Black dots are computed values of the fraction of area ice cliffs occupy within the spatial domain for each $\beta_i$ in $n$ number of iterations ($i$). The orange curve, $y(\beta)$, is a Gaussian function fit to these points (Eq. (2)). $\beta_{opt}$ is found by finding the longest distance, $d$, between $y(\beta)$ and a line passing through points $P_1$ and $P_2$ (Eq. (5) and Eq. (6)). $\beta_{opt}$ is hypothesized to be coincident with the intersection of *true positive rate* and *precision*, the optimized, best possible balance of errors. (b) *true positive rate* and *precision* (derived in Sect. 3.5) comparing method results for each $\beta_i$ to 'true' ice cliff area mapped from high resolution optical and thermal data (Sect. 2.2.1). (c-g) show the respective ice cliff maps ($A_{cliff\,i}$) for $x - axis$ ticks in (a) and (b) up to $\beta_i = 40°$, with 'true' ice cliff area ($x$) overlain for reference. (h) shows the iterative method run a final time where $\beta_i = \beta_{opt}$ and defines the automatically selected best ice cliff map. Location of Panels c-h shown in Fig. 2.



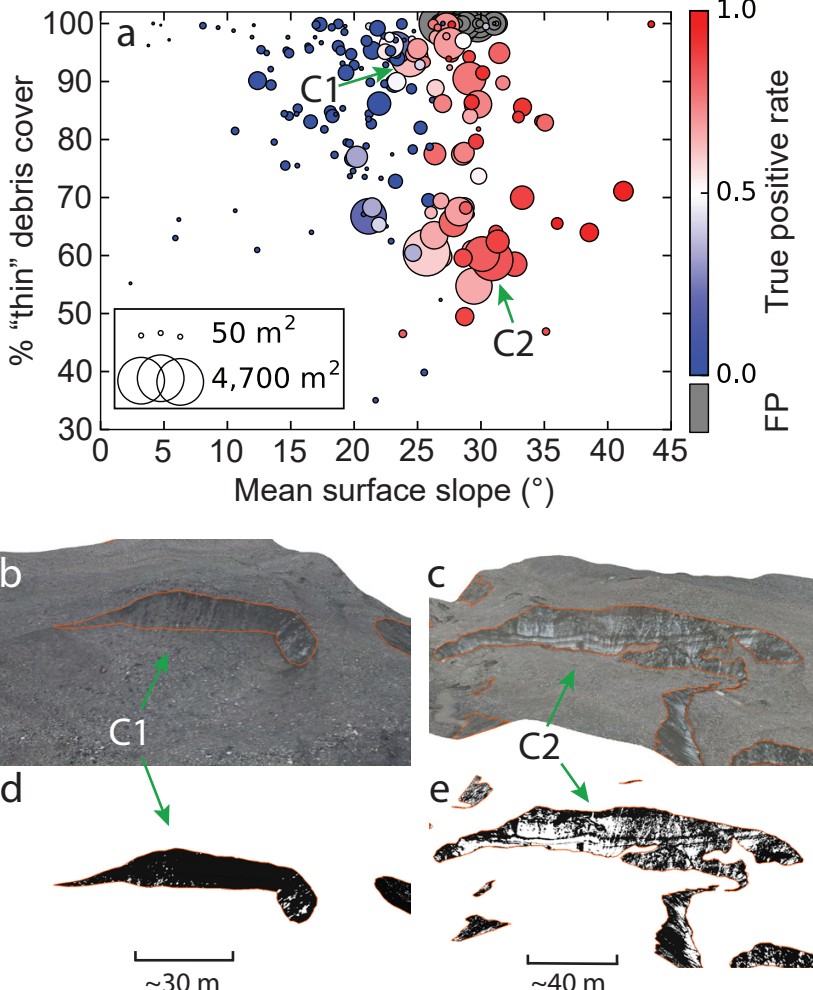

**Figure 7.** (a) each 'true' manually mapped ice cliff on the Canwell Glacier is shown as a circle sized proportionally to map view surface area and plotted against mean ice cliff surface slope and the percentage of "thin" debris cover on the ice cliff face. The color scale shows *true positive rate* from the automated ice cliff mapping method derived for each ice cliff. $FP$ ice cliffs, defined as isolated shapes that are solely $FP$ area and do not share a boundary with a 'true' ice cliff, are colored grey and abbreviated as FP on the axis label. Two ice cliffs (C1 and C2) are shown to illustrate how "thin" debris cover was mapped and provide context to the data presented in (a). (b) and (c) are oblique views of the 29 July 2016 Canwell Glacier orthomosaic with ice cliff outlines from $x$ shown in orange. (d) and (e) are the same views with the orthomosaic processed to identify only debris cover on ice cliff faces. C1 is nearly 100% debris covered which could draw into question its classification as an ice cliff. C2 is one of the more "clean" ice cliffs within the Canwell Glacier study area but is still covered by a non-trivial amount, >50%, of debris. C1 shows linear englacial debris bands that contribute to the ice cliff face debris accumulation. Location shown in Fig. 2.



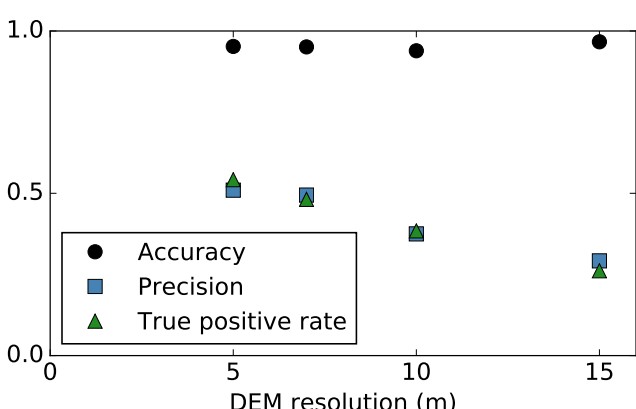

**Figure 8.** Method performance as a function of input DEM resolution.





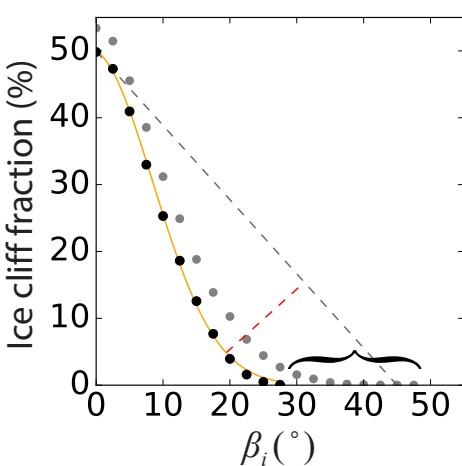

**Figure 9.** Identical to Fig. 6a, except with $TP$ and $FP$ area removed from the Canwell Glacier spatial domain to test the tool on an area that has definitively no ice cliffs. For comparison, dots shown in grey are the black dots in Fig. 6a where $TP$ and $FP$ area are not removed. The curly bracket shows the extension of the curve that is indicative of the presence of steep surface slope areas characteristic of ice cliffs. While the method still identified area (erroneously) as ice cliff, the abrupt termination of the curve with $TP$ and $FP$ area removed is an indication that there are in fact no ice cliffs present.



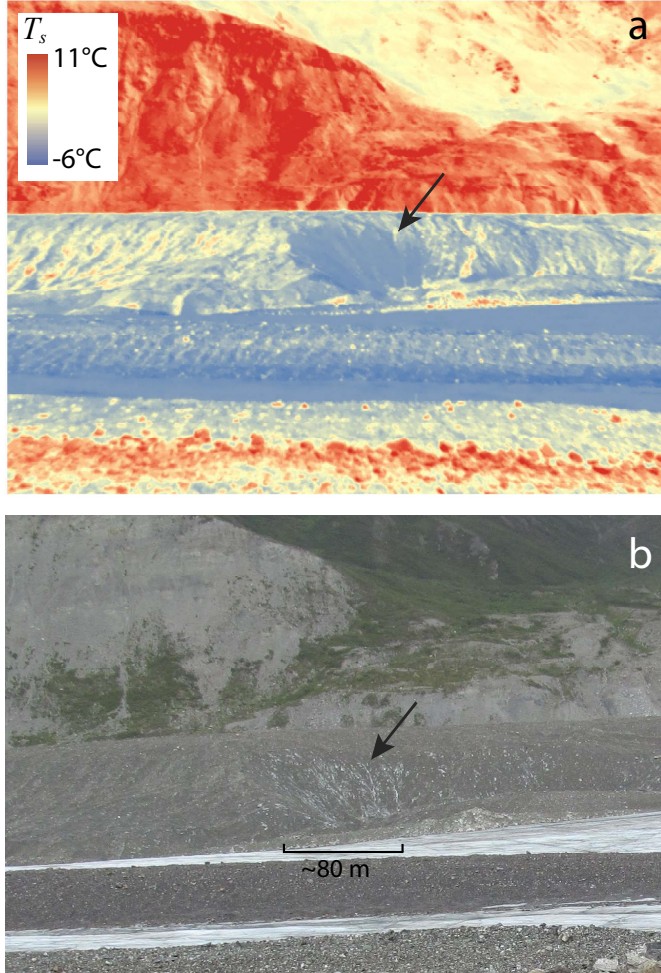

**Figure 10.** The side of a medial moraine on Canwell Glacier, possibly at an intermediate stage between being classified as an ice cliff or debris-covered area (black arrow in (a) and (b). (a) oblique image of surface temperature ($T_s$). The sharp boundary between blue and red in the middle of the image separates the top of the medial moraine and the off-glacier valley wall. (b) oblique visible image of the same feature. Location shown in Fig. 2. Bare ice $T_s$ values below 0°C are likely due to assuming a constant emissivity for the entire image.



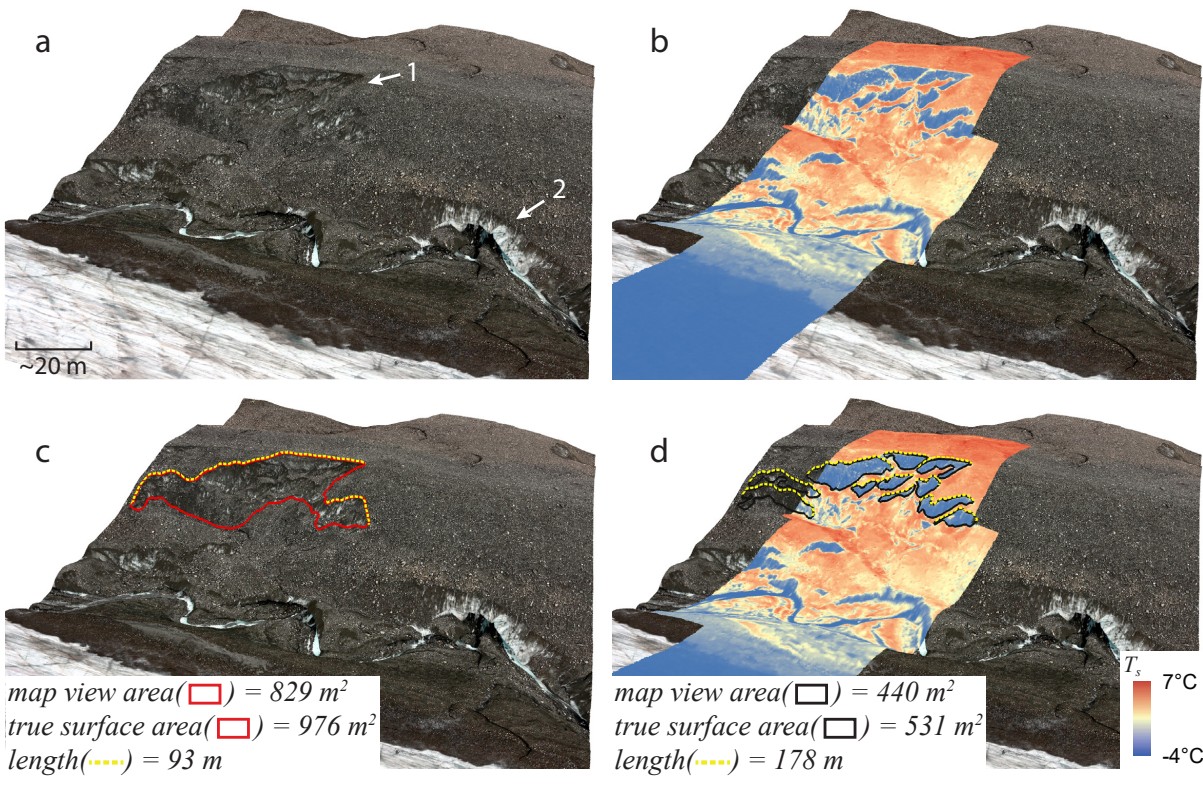

**Figure 11.** 1 and 2 in (a) show a less obvious and a very obvious ice cliff, respectively. When high resolution surface temperature ($T_s$) data is draped over ice cliff 1 in panel b, it becomes very apparent. (c) and (d) demonstrate two common methods for mapping ice cliffs: by area and by tracing the top edge. Bare ice $T_s$ values below 0°C are likely due to assuming a constant emissivity for the entire image.