# Peer review of "Automated detection of ice cliffs within supraglacial debris cover"

_The Cryosphere, 2017_

## Referee Comment (RC1) · D. Rounce (Referee) · 24 Nov 2017

**Review of "Automated detection of ice cliffs within supraglacial debris cover"**
by Sam Herreid and Francesca Pellicciotti

This study develops a new method to map ice cliffs based on the slope of a high-resolution (< 5 m) DEM. The method is developed on Canwell Glacier in Alaska and compared to ice cliffs that were delineated from high resolution visible and thermal images. The method was also applied to Ngozumpa Glacier, where a pre-existing dataset of ice cliff delineations used to assess the method's broader applicability. The developed method is quite novel in its use of a centerline extension length, which enables the method to capture the smaller ends of the ice cliffs. Another novel part of the method is the generation of probability maps and the assessment of the model's performance, which enables the accuracy and precision to be properly assessed.

For the most part, the manuscript is very well written and easy to follow. The problems associated with mapping ice cliffs are well described as is the relevance of this study. Specifically, ice cliffs on debris-covered glaciers are localized areas of high melt such that they can significantly alter the evolution of debris-covered glaciers; however, mapping ice cliffs remains difficult. The methods developed in this study are a major advance and will be a significant improvement once high resolution DEMs are available on a global scale.

The only general comments I had were concerning the use of 2D area versus 3D area and the accuracy of the validation datasets. All other comments were very minor. All in all, I believe this study is a sound contribution to the field and recommend this study for publication after minor revisions.

General Comments

One of the major improvements of this method is the ability to estimate the 3D area of the ice cliffs as opposed to 2D area typically derived from nadir-looking satellite imagery. First off, the authors refer to these areas in a variety of different ways throughout the text, e.g., true ice cliff area, area considering slope, and 3D area. While they are fairly easy to understand, it may be clearer for the reader to use one set of terminology throughout the paper. Furthermore, when comparing 2D area and 3D area (Section 4.1), the authors state the ice cliffs make up 4.9% of the map view area, but that this underrepresents the true area by 19%. This seems to imply that the true ice cliff area would be 19% greater (~6%) of the total glacier area; however, that does not factor into account what the true glacier area would be. I wonder how much the ice cliff area changes compared to the 3D glacier area? If this is substantial, then it would highlight the importance of assessing 3D area, while if it is negligible it may indicate that using 2D area is sufficient. Either way it could have important implications for modeling the evolution of debris-covered with ice cliffs included.

I found the discussion of how an ice cliff is defined to be very interesting. Specifically, determining how thin debris that is typically present on ice cliffs is considered is a challenging problem. The high resolution thermal imagery truly enabled this problem to be investigated, but I wonder how the authors "liberal outlines" influenced the ice cliff area? Is there a way to estimate the percentage of ice cliffs that were easy to include versus those that were questionable? Were the additional ice cliffs that were added to Ngozumpa Glacier all these questionable ice cliffs? If so, the percentage that was added could provide some indication as to the difference in ice cliff area that different individuals may have. Furthermore, this may enable the authors to quantify the uncertainty associated with the validation dataset, which did not

appear to be considered, i.e., this is different than the uncertainty associated with the developed method compared to the validation dataset.

Specific Comments
P1, L5: "include" not included.

P1, L14-16: Sentence is difficult to read. Perhaps, "... are still poorly understood processes, in part, due to a lack of base data, which is an obstacle for establishing a robust understanding…"

P1, L20-21: present *on* a debris *covered glacier*

P2, L16-17: "a small area of low angle… enclave" does not make sense. Both area and enclave are referring to a specific area, so it is difficult to understand.

P3, L2-3: "North and south facing ice cliffs will likely be optically distinct and crescent to circular ice cliffs will exhibit" does not make sense. Please clarify what you are trying to say.

P3, L28: "identify" not identifying

P5, L26-32: "Ngozumpa Glacier was selected for two reasons, first, …". The second reason does not appear until Line 32 making it difficult to following what the two reasons are. I would suggest either making it two separate sentences and keeping it the same, i.e., "Ngozumpa Glacier was selected for two reasons. First, …" or state the two reasons in that sentence and then go on to describe them.

P5, L26-27: this sentence is repetitive. Distinctly different geographical location and notably different from Canwell Glacier mean the same thing.

P7, Eqn 1: Why the use of Y and N as opposed to 0 and 1, which is more typical of a binary system?

P7, L23: "Where" does not need to be capitalized. Also, does the paragraph after an equation, which is still part of the previous paragraph need to be indented? If not, then change this after all equations.

P8, L8: Is the $A_{cliff,i}$/area(spatial domain) a comparison of 3D cliff area to 2D glacier area? If so, this seems as though you are comparing apples to oranges and it should be 3D area to 3D area or 2D area to 2D area.

P9, L16-17: I would recommend changing this to the positive instead of using a double negative – "it is critical that segments are large enough such that meaningful statistics can be computed".

P10, L15: Once again, ice cliff area is 3D, but is the glacier 3D area also considered in this manner? I would imagine this would impact the area associated with false positives, etc.

P11, L28: Is thin debris cover quite sensitive to this pixel value threshold? Does it greatly alter the percentage of thin debris on the ice cliffs?

P12, L6-7: How does the 19% increase alter the percentage of the total glacier area (3D in this case)? See the General Comment.

P13, L16: I would suggest stating the distance from the terminus of the lower ablation zone or state the area that was investigated based in Section 2.2.2. It's shown in Figure 1, but it may be nice to have the text as well.

P13, L24-25: Sentence is confusing. Please clarify. Specifically, what is assumed to be transferable to Ngozumpa Glacier? The model input parameters, the methods?

P14, L6: "tested" not testing

Section 5.3: The use of high-resolution thermal imagery to map ice cliffs seems to be invaluable in assessing the thin debris on ice cliffs. I am surprised that this important dataset is not mentioned with respect to future work, i.e., while higher resolution DEMs will enable this method to be applied, it appears that high resolution thermal imagery is needed to assess the accuracy of the methods in other areas, correct?

Table 2. Appear to be missing blue/best error distribution?

Table 3. No bold font as alluded to in the caption. Also, no red TP rate or blue error distribution?

Figure 1. For someone not familiar with Ngozumpa and Canwell Glaciers it may be difficult to determine which glacier is which. I would recommend stating left and right or placing (a), (b), and (c) on the figures.

Figure 3. There are 3 colors in the inset plots and yet only 2 colors in the legend. What is the third color representing?

Figure 4. May be nice to show $L_e$ on both sides.

Figure 5. The surface temperatures are counter-intuitive. Red is cold and blue is hot. Figure 10 has them as red is hot and blue is cold. I would recommend switching these such that they are intuitive and consistent.

---

## Referee Comment (RC2) · Anonymous Referee #2 · 30 Nov 2017

This manuscript tries to facilitate a more robust and operator-independent mapping of ice cliff extent on debris covered glaciers. The method is based on an investigation and classification of local surface slopes, using raster elevation maps (e.g. satellite derived DEMs). In order to assess the usefulness of such an approach, it might be worthwhile to discuss the reasons, why ice cliffs should be mapped at all. Ice cliffs are steep, smooth sections of glacier surface with no, or only a minor coverage of supra-glacial debris, embedded in glacier area with a considerably thicker debris cover. Due to this difference in debris thickness and the fact that the thin debris cover on the ice cliffs usually is below the critical thickness of the Östrem curve, these areas tend to show a strongly enhanced melt rate compared to the surrounding glacier surface. Also, the aspect and slope of ice cliffs can be favorable for melt. The crucial parameter, which

relates potential ice melt to atmospheric parameters, in this context, is the surface temperature. In fact, it is not really necessary to call something an ice cliff, if high melt rates can be derived or parameterized from other information. Unfortunately, the map view area of ice cliffs is usually much less than the available pixel resolution of thermal remote sensing products. The availability of higher resolution DEM information might therefore be a good reason to try and identify such areas of high ice loss from geometric constraints. This manuscript demonstrates a novel approach to investigate the slope distribution across debris covered glacier surfaces and relates these results to the probability of ice cliff existence. In my view, this is a considerably advance towards automated ice cliff mapping, based on the availability of remote sensing products.

Even though the authors discuss the problems connected with ice cliff definition, the main problem I see is the missing link between the classification tool and the physical conditions for ice cliff generation. The basis of the method is a threshold for surface slope, above which a slope can be considered an ice cliff. Furthermore, the probability of ice cliff occurrence is computed for a series of thresholds, producing a Gaussian distribution function of ice cliff fraction versus slope threshold. The optimum choice of slope threshold is then found as the intersection of the maximum orthogonal distance between a hypothetical line P1-P2 and the distribution function. I cannot see any physical reason why this should be a "preferred" angle in the distribution function. On page 8, line 22 it is clearly stated that this is hypothesized, but there is no attempt further in the manuscript relate that to any physical characteristic. Maybe I missed the point, but I would encourage the authors to improve this relation between the distribution function and the conditions defined as requirement for ice cliff existence. In this context, it might also be worth to discuss the choice of a Gaussian distribution function, which relates to the reasons for surface undulations on the glacier surface. Basically, it can be assumed that strain, differential melt and the existence of surface melt water are responsible for the creation of surface undulations. If these effects are randomly distributed, a Gaussian distribution function is probably a good representation. This, however, can be questioned with regards to ice cliffs, because these features are usu-

ally connected with a discontinuity in surface slope. The reason for this is that the ice cliff surfaces are not able to maintain the original debris cover and melt rates change abruptly at the cliff boundaries, leading to characteristic cliff slope angles.

Besides this lacking linkage between the statistical model and the physical world, this manuscript is a valuable contribution to the important issue of including ice cliffs in the mass balance estimates of debris covered glaciers. I add minor comments, according to their appearance in the manuscript: P.2,L.7: What do you mean with "conventional input data". This should be a bit more elaborate.

P. 2,L.25: I do not see that this reduction of D is correct. The ice cliff transect s oriented parallel to x, but D is the ice cliff width from bottom to top, which is in an angle (perpendicular) to x. In this case an integral 0-D by dx makes no sense.

P.3,L.8/9: Could you clarify what ambiguities you mean? Your method only aims on slope, while radiometric sensors aim on roughness, brightness and temperature. There might be a range of possible ambiguities.

P.3,L.6-20: This paragraph is rather unclear to me. Is there any relation to published observations? Based on my observations (of course depending on the definition of an ice cliff), cliff have no ability to accumulate any debris larger than small rock flakes. The temporal evolution of cliffs shows that coarse debris can accumulate at the bottom and slowly covers the cliff, but then it should not be considered a cliff anymore. As I mentioned in the introduction, in my opinion the definition of an ice cliff makes only sense if it is connected to the considerable difference in thermal fluxes. Therefore the geometry aspect is only a supporting approach.

P.3,L.24: Cliff slopes mainly differ due to aspect and thus solar radiation. There are several publications connected to this topic. It is likely that there is some overlap, but it is probably rather small, because what could be the physical reason that coarse debris sticks to one part of the slope, but it slides from the other part with the same angle? The only situation I can see is a small and short slope, where the talus is large enough

to prevent additional mass movement from the steeper part.

P.3,L.29: The spatial resolution below 1m seems a random choice without any example from reality. You could provide some real observations about typical ice cliff height, slope and map view expression, to demonstrate which resolution is required to clearly capture ice cliffs.

P.3,L.31: Again the definition of conventional is missing.

P.3,L.32: 5m might be moderate for visible imagery, but for large coverage elevation data this is still high resolution.

P.4,L.23: High vertical accuracy is not critical for cliff localization, but for correct slope calculation, the relative accuracy is decisive.

P.5,L.2: Can you shortly specify the data used for calibration already here?

P.5,L.7: What do you mean with different surfaces? Types?

P.5,L.13: can you provide some specifications about the data collection? Chip dimensions, mean flight elevation, ground resolution, spatial overlap?

P.5,L.16: Again, can you please provide the spatial ground resolution?

P.5,L.30-32: This sentence is difficult to understand. Do you mean the differences in the ice cliff slope distribution indicates that a unique value cannot be used for larger regions?

P.6,L.10: Is this an additional GeoEye scene (23rd December), compared to the one used before (29th December)? Please clarify.

P.6,L.23: what is an "area threshold"?

P.6,L.27: As in many other cases, the manuscript would be more easily readable if expressions could be simplified: "elevation difference" instead of "rate of change in elevation". Please check also other cumbersome expressions.

P.7,L.21: what does "piecewise" mean in this context? This description is misleading. The iterative process is based on n iterations of varying beta. But the probability model is not "piecewise".

P.8,L.6: The parameter of the probability map is p(x) not beta. Maybe it could be written: ice cliff probability maps p_i(x) in dependence of beta_i.

P.8,L.7: see comment above about complicating the readability: "vector ice cliff cape area" basically means "the resulting ice cliff area"

P.8,L.11/12: This sentence does not explain the characteristics of the function in relation to the parameters: Why are the y(beta) are unrealistically high for low beta? y'(beta) approaches 0 for larger betas due to the nature of the exponential function. This is true for the existence of ice cliffs, but also without. It is rather a distinction that high betas do not occur, if there are no cliffs, because debris cannot be maintained on steep slopes.

P.8,L.16: The formulation most accurate final A_cliff and coupled p(x) is not necessarily true. It is probably the optimum combination of A_cliff and p(x).

P.8,L.22: please refer to Fig. 6 already here, so that the reader can relate this difficult description to the graphical expression.

P.8, eq.4: what mathematical form should "distance (P1,P2,(beta, y(beta))) represent"? This formulation seems strange. Can you give some indication how you came to the resulting right hand side of eq. 4?

P.9;l.5: It should already be mentioned here that gama has small thresholds.

P.9,L.6: I do not see any problem with also calculating realistic values for P2 at beta=90°. Maybe it would be a better approach to use the point of inflexion of the gamma function as optimum. Because this is defined, based on the approximated parameters a, b and c only.

P9,L.10: This is not an additional iteration, but a final application of the model with the found beta_opt.

P9.,L.11/12: Does this indicate that you just use a manual beta_i if the methods fails? What would be the potential reasons for the method failing?

P.9,L. 20: The fractional debris covered area represents the original classification of debris covered glacier, where embedded clean ice is included in the debris cover?

P.10,L.6: The truth dataset x is probably different from the parameter x in p(x). Can you explain?

P.10,L.15: What is the secant of a slope pixel? Here it seems that you calculate a volume (length by area) instead of an area (length by length). Please reformulate.

P.10.L.19/20: Again please simplify: "tool output vector shape defined by A_cliff" means probably the same as "A_cliff vector shapes".

P.11,L.12: delete "for error distribution <1".

P.12,L.2: Can you comment here on the color coded distribution of "true positive" results in fig. 7? It seems there is a clear distinction in dependence of beta. What is the meaning of that?

P.12,L.9: magnitude (typo)

P.12,L.19: What would be the difference in this example of Fig. 6, using P_1 (beta_i) for 60° and 90°?

P.12,L.20/21: this needs more emphasis, that the truly artificial determination of beta_opt from the Gaussian distribution matches with the error optimum based on the calibration.

P.13,L.14/15: I do not understand this sentence. Can you please elaborate on the limitations with respect to slope angle?

P.13,L.14: Low slope means that the spatial dimension of the cliff is so narrow that the elevation difference between pixels is dominated by the more flat debris covered parts. The cliff portion itself is still "steep". Beyond the ice cliff means, beyond the spatial classification result.

P.13,L.19: What means "surface slope may be saturated"?

P.15,L.11: The example of Figure 8 should be accompanied by the characteristics of the ice cliff results from Canwell Glacier, because the performance depends clearly on the typical ice cliff width and slope distribution.

P.15,L.17: There still exists a result for beta_opt: about $19°$ and y_opt: about 4%. The only indication that the method fails is the non-assymptotic shape of the fuction towards the x-axis? What is the reasoning behind this?

P.15,L.29: What is meant with "mass wasting"? Is it removal of supra-glacial debris? Mass wasting usually is used for ice loss.

Fig. 5: Panel (e) is not mentioned on the caption.

---

## Author Comment (AC1) · 22 Jan 2018

**Response to referee comments regarding Herreid and Pellicciotti, "Automated detection of ice cliffs within supraglacial debris cover"**

https://doi.org/10.5194/tc-2017-151

January 22, 2018

**Comments from David Rounce (RC1)**

We would like to thank David Rounce for his careful, detailed and very helpful comments. His point regarding the 3D consideration of the entire spatial domain was improved the quality of our results. We first address his general comments and then respond, point by point, to his specific comments. Reviewer comments are indented with a grey bar with our response below.

> This study develops a new method to map ice cliffs based on the slope
> of a highresolution (< 5 m) DEM. The method is developed on Canwell
> Glacier in Alaska and compared to ice cliffs that were delineated
> from high resolution visible and thermal images.  The method was also
> applied to Ngozumpa Glacier, where a pre-existing dataset of ice cliff
> delineations used to assess the method's broader applicability.  The
> developed method is quite novel in its use of a centerline extension
> length, which enables the method to capture the smaller ends of the
> ice cliffs.  Another novel part of the method is the generation of
> probability maps and the assessment of the model's performance, which
> enables the accuracy and precision to be properly assessed.  For the
> most part, the manuscript is very well written and easy to follow.
> The problems associated with mapping ice cliffs are well described
> as is the relevance of this study.  Specifically, ice cliffs on
> debris-covered glaciers are localized areas of high melt such that
> they can significantly alter the evolution of debris-covered glaciers;
> however, mapping ice cliffs remains difficult.  The methods developed
> in this study are a major advance and will be a significant improvement
> once high resolution DEMs are available on a global scale.  The only
> general comments I had were concerning the use of 2D area versus 3D area
> and the accuracy of the validation datasets.  All other comments were
> very minor.  All in all, I believe this study is a sound contribution
> to the field and recommend this study for publication after minor
> revisions.

> One of the major improvements of this method is the ability to estimate
> the 3D area of the ice cliffs as opposed to 2D area typically derived
> from nadir-looking satellite imagery.  First off, the authors refer to
> these areas in a variety of different ways throughout the text, e.g.,
> true ice cliff area, area considering slope, and 3D area.  While they
> are fairly easy to understand, it may be clearer for the reader to use
> one set of terminology throughout the paper.

We agree that several terms are used to describe the same quantity (true ice cliff surface area, area considering slope, 3D surface) but we think that depending on the context each makes sense, e.g. if discussing a comparison to map-view surface area, it is the most intuitive to use 'true surface area' while when discussing the various methods for ice cliff mapping, it seems best to talk in 3D, 2D, 1D terms. At the first mention of a 3D surface (Section 3.2.1) we have added "...are converted to 3D surfaces (...used synonymously with 'true surface area' and 'area considering slope')." and at many location where we say 'true surface area' we in parenthesis say "area considering slope". If the Reviewer or Editor disagree and find this to be unnecessary confusion we are happy to select one and be consistent throughout.

> Furthermore, when comparing 2D area and 3D area (Section 4.1), the
> authors state the ice cliffs make up 4.9% of the map view area, but that
> this underrepresents the true area by 19%.  This seems to imply that
> the true ice cliff area would be 19% greater ( 6%) of the total glacier
> area; however, that does not factor into account what the true glacier
> area would be.  I wonder how much the ice cliff area changes compared
> to the 3D glacier area?  If this is substantial, then it would highlight
> the importance of assessing 3D area, while if it is negligible it may
> indicate that using 2D area is sufficient.  Either way it could have
> important implications for modeling the evolution of debriscovered with
> ice cliffs included.

This is a very useful comment, and the reviewer is right that there are differences between the 2D and 3D areas, which we have neglected to consider. We have now calculated the 3D area and updated Section 4.1 to included 3D glacier area. There is a difference of the percentage of ice cliffs within the study area from 4.9% in map view to 5.7% considering slope.

> I found the discussion of how an ice cliff is defined to be very
> interesting.  Specifically, determining how thin debris that is
> typically present on ice cliffs is considered is a challenging
> problem.  The high resolution thermal imagery truly enabled this
> problem to be investigated, but I wonder how the authors ''liberal
> outlines'' influenced the ice cliff area?  Is there a way to estimate
> the percentage of ice cliffs that were easy to include versus those
> that were questionable?  Were the additional ice cliffs that were
> added to Ngozumpa Glacier all these questionable ice cliffs?  If so,
> the percentage that was added could provide some indication as to
> the difference in ice cliff area that different individuals may have.
> Furthermore, this may enable the authors to quantify the uncertainty
> associated with the validation dataset, which did not appear to be
> considered, i.e., this is different than the uncertainty associated with
> the developed method compared to the validation dataset.

There are several good points and questions within this comment. Subjectivity is an often under quantified variable in manually generated data. A common approach to quantifying subjectivity is having several technicians conduct the same mapping task independently and compare the variability between technicians. While this is a worthwhile method, it is still limited to the data the technicians are given and their personal mapping criteria which could both contribute to a converging result that is still not 'true'. For this study we tried a different approach to minimizing subjectivity by collecting corroborating datasets. By using high resolution visible and thermal data we were able to cross validate our ice cliff outlines. However, even with these data we found the manual mapping difficult as there was a continuous spectrum of ice cliffs between easily identified to questionable identification. Because of this spectrum we don't know of a robust way to delimit ice cliffs between easy and questionable identification. To make this distinction clear we added to Section 3.4: "While we made an effort to manually map ice cliffs based on consistent criteria, there is subjective interpretation within this 'truth' dataset. We did not quantify the uncertainty associated with this subjectivity." Regarding Ngozumpa Glacier, our preference would have been to make no additions to the ice cliff map, yet we also thought that the mapping criteria should be consistent with how we mapped ice cliffs for Canwell Glacier. Of the ice cliff additions we made, 9 had what we would call a considerable "thin" debris covering, 7 were in dark shadow making a very confident classification difficult, 4 were what we would interpret as omission errors and several of these and around 13 others were quite small. Because of the variable reasons for omission, it is not clear to us what can be considered questionable. If two technicians independently zoom in to a region and agree there is a small ice cliff but for further mapping one technician maps from only half the zoom and no longer finds small ice cliffs while the other does, are the missed small ice cliffs questionable or not questionable? We very much like the the questions raised but did not attempt to quantify the inherent subjectivity.

**Specific comments**

> P1, L5:  ''include'' not included.

Correction made.

> P1, L14-16: Sentence is difficult to read. Perhaps, ''... are still
> poorly understood processes, in part, due to a lack of base data, which
> is an obstacle for establishing a robust understanding...''

Changed as suggested with an additional shortening of the sentence.

> P1, L20-21: present on a debris covered glacier

Changed as suggested.

> P2, L16-17: ''a small area of low angle... enclave'' does not make
> sense. Both area and enclave are referring to a specific area, so it is
> difficult to understand.

Changed to "...a small area of low angle (i.e. not cliff) bare glacier ice located within a debris-covered portion of a glacier..."

> P3, L2-3: ''North and south facing ice cliffs will likely be optically
> distinct and crescent to circular ice cliffs will exhibit'' does not
> make sense. Please clarify what you are trying to say.

Changed to: "North and south facing ice cliffs present in a single image will likely appear either dark (shaded bare ice) or light (illuminated bare ice) relative to unshaded surrounding debris cover. Crescent to circular ice cliffs will likely exhibit a spectrum of shade and illumination."

> P3, L28: ''identify'' not identifying

This sentence has been removed in response to a comment below.

> P5, L26-32: ''Ngozumpa Glacier was selected for two reasons, first,
> ...''. The second reason does not appear until Line 32 making it
> difficult to following what the two reasons are. I would suggest
> either making it two separate sentences and keeping it the same, i.e.,
> \Ngozumpa Glacier was selected for two reasons. First, ..." or state the
> two reasons in that sentence and then go on to describe them

Changed as suggested: "Ngozumpa Glacier was selected to be used in this study for two reasons. First, it is".

> P5, L26-27: this sentence is repetitive. Distinctly different
> geographical location and notably different from Canwell Glacier mean
> the same thing.

We agree the initial wording was repetitive. We have added more details to clarify our point: "First, it is located in a geographical region that is different from the Alaska Range with respect

to latitude, longitude, continentality, climate and orogeny. These factors and others establish a setting where exposure to the sun and the overall debris cover (e.g. debris extent, clast sizes, and thickness) at Ngozumpa Glaicer is notably different from the Canwell Glacier."

> P7, Eqn 1:  Why the use of Y and N as opposed to 0 and 1, which is more
> typical of a binary system?

Changed as suggested.

> P7, L23:  \Where" does not need to be capitalized.  Also, does the
> paragraph after an equation, which is still part of the previous
> paragraph need to be indented?  If not, then change this after all
> equations.

With particular respect to P7 L23 we ended the sentence with Eq. 1 to avoid a very run on sentence, but are happy to change this at the Editor's request. We corrected the indentations here (P7 L23) and throughout as suggested.

> P8, L8:  Is the Acliff,i/area(spatial domain) a comparison of 3D cliff
> area to 2D glacier area?  If so, this seems as though you are comparing
> apples to oranges and it should be 3D area to 3D area or 2D area to 2D
> area.

The quantity Acliff,i/area(spatial domain) described at P8, L8 is a ratio of two 2D areas. To help eliminate confusion we have added the sentence at the beginning of Section 3.2.1: " "This area, and all subsequent areas derived in the mapping method, are simplified as a shape in two dimensions (2D). Only final results of the mapping method are converted to 3D surfaces (Sect. 4 and 5.3, method described in Sect. 3.4)."

> P9, L16-17:  I would recommend changing this to the positive instead
> of using a double negative { ''it is critical that segments are large
> enough such that meaningful statistics can be computed''.

Changed as suggested.

> P10, L15:  Once again, ice cliff area is 3D, but is the glacier 3D area
> also considered in this manner?  I would imagine this would impact the
> area associated with false positives, etc.

For the statistical measures of performance analysis we consider only 2D area. 3D true surface area is only used when presenting final (more science question oriented) results. The majority of the article is focused on mapping and a consistent use of 2D surface area simplifies the problem. We have made this more explicit in P10 L15: "True surface area (as opposed to map view surface area) was calculated to present final results (Sect. 4 and 5.3) by multiplying..."

> ```
> P11, L28:  Is thin debris cover quite sensitive to this pixel value
> threshold?  Does it greatly alter the percentage of thin debris on the
> ice cliffs?
> ```

We did not conduct a quantitative sensitivity analysis and have added this caveat to the text: "We did not quantify the sensitivity of this threshold parameter and relied on visual assessment (e.g. comparing panels b and c to panels d and e in Fig. 7) to determine a single values that minimized errors (e.g. due to varying lithology)and maximize success."

> ```
> P12, L6-7:  How does the 19% increase alter the percentage of the total
> glacier area (3D in this case)?  See the General Comment
> ```

The text has been updated with the following new information: "The manually generated, 'true' ice cliff dataset shows 4.9% of the 1.74 km$^2$ Canwell Glacier study area is ice cliff in map view (not considering slope). Ice cliff map view area (84,630 m$^2$) under represents true ice cliff surface area (104,920 m$^2$, considering slope) by 19%. Considering the true surface area of the Canwell Glacier study area, map view under represents true surface area (1.86 km$^2$) by 6%. Considering both true surface area of ice cliffs and the Canwell Glacier study area, the fraction of ice cliff area is 5.7%."

> ```
> P13, L16:  I would suggest stating the distance from the terminus of
> the lower ablation zone or state the area that was investigated based in
> Section 2.2.2.  It's shown in Figure 1, but it may be nice to have the
> text as well.
> ```

Changed as suggested.

> ```
> P13, L24-25:  Sentence is confusing.  Please clarify.  Specifically,
> what is assumed to be transferable to Ngozumpa Glacier?  The model input
> parameters, the methods?
> ```

Changed to: "Considering that not a single alteration was made to the method and input parameter values used for Canwell Glacier, the method mitigated the many physical differences between Canwell and Ngozumpa glaciers (described in Sect. 2.2.2 and shown in Fig. 3) as well as different DEM generation methods (satellite based rather than airborne structure from motion). This ability to accommodate different physical characteristics enables the method to outperform a simple slope threshold found at one location, e.g. at Canwell Glacier, and assume that it is transferable to other places on Earth, e.g. Ngozumpa Glacier (Tables 2 and 3)"

> ```
> P14, L6:  \tested" not testing
> ```

Correction made.

> Section 5.3:  The use of high-resolution thermal imagery to map ice
> cliffs seems to be invaluable in assessing the thin debris on ice cliffs.
> I am surprised that this important dataset is not mentioned with respect
> to future work, i.e., while higher resolution DEMs will enable this
> method to be applied, it appears that high resolution thermal imagery
> is needed to assess the accuracy of the methods in other areas, correct?

We believe this is correct but were also hesitant to be too explicit when making suggestions to future scientists, perhaps there is even better validation data that we have not thought of. We altered a sentence in Section 6 to: "Further validation of this method using ice cliff maps from high resolution visible, thermal or other data in other regions will help to either support or discredit our claim of wide applicability."

> Table 2.  Appear to be missing blue/best error distribution?

Correction made.

> Table 3.  No bold font as alluded to in the caption.  Also, no red TP
> rate or blue error distribution?

Correction made, with additional cells added to clarify now more precise corresponding text in Section 4.2.

> Figure 1.  For someone not familiar with Ngozumpa and Canwell Glaciers
> it may be difficult to determine which glacier is which.  I would
> recommend stating left and right or placing (a), (b), and (c) on the
> figures.

Changed as suggested with left and right in the caption.

> Figure 3.  There are 3 colors in the inset plots and yet only 2 colors
> in the legend.  What is the third color representing?

Changed as suggested.

> Sentnece added to the caption of Figure 3:  ''The grey/orange region
> of overlap illustrates the difficulty of using surface slope alone to
> identify ice cliffs.''

Changed as suggested.

> Figure 4.  May be nice to show Le on both sides

We think it is sufficiently clear with one label given the color difference and the legend directly above, but are happy to add this at the Editor's request.

> Figure 5. The surface temperatures are counter-intuitive. Red is cold
> and blue is hot. Figure 10 has them as red is hot and blue is cold.
> I would recommend switching these such that they are intuitive and
> consistent.

This was an intentional deviation from consistency and intuition, now explained in the caption: "The color gradient is flipped so that cold ice cliffs easily stand out as red can be easily compared with [panel] (d).". We can still change this if the reviewer/editor is not convinced by our argument.

**Comments from Anonymous (RC2)**

We thank Reviewer 2 for her/his careful, detailed and helpful comments. We were especially appreciative of the close attention to mathematical details. We first address her/his general comments and then respond, point by point, to her/his specific comments.

> This manuscript tries to facilitate a more robust and
> operator-independent mapping of ice cliff extent on debris covered
> glaciers. The method is based on an investigation and classification
> of local surface slopes, using raster elevation maps (e.g. satellite
> derived DEMs). In order to assess the usefulness of such an approach,
> it might be worthwhile to discuss the reasons, why ice cliffs should
> be mapped at all. Ice cliffs are steep, smooth sections of glacier
> surface with no, or only a minor coverage of supra-glacial debris,
> embedded in glacier area with a considerably thicker debris cover.
> Due to this difference in debris thickness and the fact that the thin
> debris cover on the ice cliffs usually is below the critical thickness
> of the Östrem curve, these areas tend to show a strongly enhanced
> melt rate compared to the surrounding glacier surface. Also, the
> aspect and slope of ice cliffs can be favorable for melt. The crucial
> parameter, which relates potential ice melt to atmospheric parameters,
> in this context, is the surface temperature. In fact, it is not really
> necessary to call something an ice cliff, if high melt rates can be
> derived or parameterized from other information. Unfortunately, the map
> view area of ice cliffs is usually much less than the available pixel
> resolution of thermal remote sensing products. The availability of
> higher resolution DEM information might therefore be a good reason to
> try and identify such areas of high ice loss from geometric constraints.

We completely agree that there may be other approaches or datasets that could make the mapping of ice cliffs (especially through geometric constraints) obsolete. However, as the reviewer points out, at the present time more favorable datasets have not yet become available or have not yet been exploited/introduced into the debris covered glacier research community. While something more robust will surly emerge it is with the goal of addressing large scale problems now that motivated the writing of this paper. We added to the second sentence of the introduction a stated link between resolving where ice cliffs are with respect to solving for glacier melt in heavily debris-covered regions.

This manuscript demonstrates a novel approach to investigate the slope
distribution across debris covered glacier surfaces and relates these
results to the probability of ice cliff existence.  In my view, this
is a considerably advance towards automated ice cliff mapping, based on
the availability of remote sensing products.  Even though the authors
discuss the problems connected with ice cliff definition, the main
problem I see is the missing link between the classification tool and
the physical conditions for ice cliff generation.  The basis of the
method is a threshold for surface slope, above which a slope can be
considered an ice cliff.  Furthermore, the probability of ice cliff
occurrence is computed for a series of thresholds, producing a Gaussian
distribution function of ice cliff fraction versus slope threshold.  The
optimum choice of slope threshold is then found as the intersection of
the maximum orthogonal distance between a hypothetical line P1-P2 and
the distribution function.  I cannot see any physical reason why this
should be a ''preferred'' angle in the distribution function.  On page
8, line 22 it is clearly stated that this is hypothesized, but there
is no attempt further in the manuscript relate that to any physical
characteristic.  Maybe I missed the point, but I would encourage the
authors to improve this relation between the distribution function
and the conditions defined as requirement for ice cliff existence.
In this context, it might also be worth to discuss the choice of a
Gaussian distribution function, which relates to the reasons for surface
undulations on the glacier surface.  Basically, it can be assumed that
strain, differential melt and the existence of surface melt water
are responsible for the creation of surface undulations.  If these
effects are randomly distributed, a Gaussian distribution function is
probably a good representation.  This, however, can be questioned with
regards to ice cliffs, because these features are usually connected
with a discontinuity in surface slope.  The reason for this is that
the ice cliff surfaces are not able to maintain the original debris
cover and melt rates change abruptly at the cliff boundaries, leading
to characteristic cliff slope angles.  Besides this lacking linkage
between the statistical model and the physical world, this manuscript is
a valuable contribution to the important issue of including ice cliffs
in the mass balance estimates of debris covered glaciers

Firstly, we made a semantic error that misled the reviewer.  While the shape of Eq. 2 is
that of a Gaussian distribution, we were not using it to model a distribution, we were simply
fitting a model to data and the shape of Eq. 2 fit the data well. Since the single objective
of this study was mapping existing ice cliffs, a discussion of physical conditions for ice cliff
generation is out of the scope of this work, and reflects the fact that, despite major recent
progresses in our understanding of ice cliffs over debris covered glaciers, we still know too little
about cliff generation in particular. We do thank the reviewer for pushing us to consider the
physical reality that underpins hypothetical lines and "preferred" angles. At P.8 L.22 [location
in original submission] we tried to fully describe the hypothesized relation of an optimum slope
threshold with physical characteristics:"The hypothesis is that as $\beta_i$ increases and less sloped
debris-covered area (e.g. from strain and differential melt) is included as mapped ice cliff, true
ice cliffs will begin to comprise the majority of the mapped ice cliff fraction ($y_i$). We hypothesize

this to have a stabilizing effect (ice cliffs are a small, consistently steep area), lowering the rate of loss of mapped ice cliff area and slope of the curve as $\beta_i$ continues towards 90°. We hypothesize that this so-called 'mid' point or 'elbow' will thereby identify a $\beta_{opt}$ that reflects a surface slope characteristic common to the small spatial domain the method is applied to but possibly unique to a geographical region or latitude." Further additions emphasizing the relation between physical characteristics and our methodological choices are described per specific comments from Reviewer 2 below.

**Specific comments**

> P.2,L.7:  What do you mean with ''conventional input data''.  This
> should be a bit more elaborate.

By "conventional" we meant what the second half of the sentence describes. We have restructured the sentence to be more clear: "...requires input data that currently are, or are starting to become, freely available globally (hereafter referred to as 'conventional' data)"

> P. 2,L.25:  I do not see that this reduction of D is correct.  The ice
> cliff transect s oriented parallel to x, but D is the ice cliff width
> from bottom to top, which is in an angle (perpendicular) to x.  In this
> case an integral 0-D by dx makes no sense

We believe we have corrected this error by defining D as the map view transect distance rather than the true distance in x,z space, this should allow a meaningful integration with respect to x: "a nadir looking sensor will capture the width of an ice cliff in map-view as, $D$, a distance that under represents the true distance from the bottom debris-ice interface to the top debris-ice interface by a factor of $\int_0^D cos(\beta)dx$, where $\beta$ is surface slope along the ice cliff transect, oriented parallel to the $x$-axis."

> P.3,L.8/9:  Could you clarify what ambiguities you mean?  Your method
> only aims on slope, while radiometric sensors aim on roughness,
> brightness and temperature.  There might be a range of possible
> ambiguities.

We have reworded the leading sentence for clarity and added a detailed example of a specific ambiguity: "A "thin" debris layer is undetectable from DEM data (with the possible exception of data with a spatial resolution that is sufficiently below the size of the rock clasts/fragments). With data at a sufficient resolution (dependent on clast size and abundance), "thin" debris can be detected by visible spectrum or thermal sensors which can both facilitate mapping this quantity but also possibly introduce ambiguities when defining ice cliff area. For example, if the same ice cliff is mapped from thermal data twice in the same (summertime) day, once at night (where, for this example, the "thin" debris is $< 0°$C, in thermal equilibrium with the neighboring bare ice and is thus undetectable) and again midday (where the "thin" debris is the same, $> 0°$C, temperature as the debris cover surrounding the ice cliff and classified as such), a single scientist might generate two very different maps of an ice cliff area that, in reality, experienced no significant chance."

P.3,L.6-20:  This paragraph is rather unclear to me.  Is there any
relation to published observations?  Based on my observations (of course
depending on the definition of an ice cliff), cliff have no ability
to accumulate any debris larger than small rock flakes.  The temporal
evolution of cliffs shows that coarse debris can accumulate at the
bottom and slowly covers the cliff, but then it should not be considered
a cliff anymore.  As I mentioned in the introduction, in my opinion
the definition of an ice cliff makes only sense if it is connected to
the considerable difference in thermal fluxes.  Therefore the geometry
aspect is only a supporting approach.

We acknowledge that this paragraph might not have global applicability and have changed the leading sentence to reflect this: "A factor that may be abundant in some regions and add to the complexity of identifying and mapping ice cliffs is the presence of thin or sparse debris cover on an ice cliff face...". With regard to whether this paragraph should be included at all, we point to Figures 3,5,7b,c,10 and 11 of this manuscript where debris cover on ice cliff faces on Canwell Glacier in Alaska is, we think, quite clear visually and quantified in Figure 7a. To our knowledge there are currently no published studies that discuss in detail ice cliffs in Alaska or North America. We hint at the question of whether abundant "thin" debris cover is specific to some regions, e.g. p.5 L.28 (location in original manuscript submission), but the focus of this article is mapping ice cliffs and the associated ambiguities. We leave this possibly interesting question of relative abundances of "thin" debris cover in different geographical regions to a future study.

P.3,L.24:  Cliff slopes mainly differ due to aspect and thus solar
radiation.  There are several publications connected to this topic.  It
is likely that there is some overlap, but it is probably rather small,
because what could be the physical reason that coarse debris sticks to
one part of the slope, but it slides from the other part with the same
angle?  The only situation I can see is a small and short slope, where
the talus is large enough to prevent additional mass movement from the
steeper part.

We agree that a physical explanation of why there is overlap is lacking in this (mapping focused) manuscript and from our investigation. Any explanation would be speculative without a deeper process based study. However, both insert plots in Figure 3 show substantial overlap. Because the overlapping area (1) is substantial; (2) appears in DEM data from two different data acquisition sources; and (3) considerable care went into manually digitizing ice cliff area, we were convinced that the overlap is real and not entirely composed of error and/or inaccurate data.

P.3,L.29:  The spatial resolution below 1m seems a random choice without
any example from reality.  You could provide some real observations
about typical ice cliff height, slope and map view expression, to
demonstrate which resolution is required to clearly capture ice cliffs.

We have rewritten the sentence and hope it adds some of the desired clarity: "While very high ¡1 m spatial resolution data capable of not only resolving ice cliff faces but clasts of surrounding

debris would be ideal for mapping confidence, data at this resolution are neither freely available or available at large scales (e.g. for a whole mountain range)."

> P.3,L.31:   Again the definition of conventional is missing.

Addressed in comment above and now defined on P.2 L.7.

> P.3,L.32:   5m might be moderate for visible imagery, but for large
> coverage elevation data this is still high resolution.

We have removed the word 'moderate' here and throughout the manuscript.

> P.4,L.23:   High vertical accuracy is not critical for cliff localization,
> but for correct slope calculation, the relative accuracy is decisive.

Changed as suggested.

> P.5,L.2:   Can you shortly specify the data used for calibration already
> here?

Changed as suggested.

> P.5,L.7:   What do you mean with different surfaces?   Types?

Changed as suggested.

> P.5,L.13:   can you provide some specifications about the data
> collection?   Chip dimensions, mean flight elevation, ground resolution,
> spatial overlap?

Additions made as suggested with the exception of spatial overlap which is now mentioned but not quantified and we think that camera manufacturer and model are sufficient details for the purpose of this study, further camera specifications are easily found with this information. However if the Editor prefers these specifications be included we are happy to add them.

> P.5,L.16:   Again, can you please provide the spatial ground resolution?

Changed as suggested.

> P.5,L.30-32:   This sentence is difficult to understand.   Do you mean the
> differences in the ice cliff slope distribution indicates that a unique
> value cannot be used for larger regions?

Changed to: "The shift between the two ice cliff slope distributions shown in the two inset histograms of Fig. 3 suggest that even if overlooking overlap errors, a simple surface slope threshold deemed suitable to define ice cliffs at one location may capture a different portion of the distribution in other regions on Earth."

> P.6,L.10:  Is this an additional GeoEye scene (23rd December), compared
> to the one used before (29th December)?  Please clarify.

Corrected. Thank you for catching this error.

> P.6,L.23:  what is an \area threshold"?

The sentence in question ("Bare ice below the area threshold (included as debris-covered area) that is not part of an ice cliff will be rejected as ice cliff area in the subsequent step based on low surface slopes.") refers to the method mitigating patches of flat bare ice that are similar size to an ice cliff and surrounded by debris cover. This is probably a very minor nuance/technicality in the problem of mapping ice cliffs and we have removed the entire sentence.

> P.6,L.27:  As in many other cases, the manuscript would be more easily
> readable if expressions could be simplified:  ''elevation difference''
> instead of ''rate of change in elevation''.  Please check also other
> cumbersome expressions

Changed as suggested.

> P.7,L.21:  what does ''piecewise'' mean in this context?  This
> description is misleading.  The iterative process is based on n
> iterations of varying beta.  But the probability model is not
> ''piecewise''

We have removed the word "piecewise".

> P.8,L.6:  The parameter of the probability map is p(x) not beta.  Maybe
> it could be written:  ice cliff probability maps $p_i(x)$ in dependence of
> $\beta_i$.

Removed "$(\beta_i)$".

> P.8,L.7:  see comment above about complicating the readability:  \vector
> ice cliff cape area" basically means ''the resulting ice cliff area''

Changed as suggested.

```
P.8,L.11/12:  This sentence does not explain the characteristics of
the function in relation to the parameters:  Why are the y(beta) are
unrealistically high for low beta?  y'(beta) approaches 0 for larger
betas due to the nature of the exponential function.  This is true
for the existence of ice cliffs, but also without.  It is rather a
distinction that high betas do not occur, if there are no cliffs,
because debris cannot be maintained on steep slopes
```

This section has been rewritten to better relate the method described to the physical setting. "The curve expresses unrealistically high $y(\beta)$ with low values of $\beta$ because the threshold slope for ice cliff classification is well below slopes from surface roughness/undulations common for a debris-covered portion of a glacier. If there are ice cliffs within the spatial domain, the increase in $\beta$ towards 90° causes the threshold to become too stringent, excluding even true ice cliff area. Steep ice cliff faces (and possibly erroneous DEM data) will cause iterations to run through high values of $\beta$ with minor reductions in $y(\beta)$ causing the slope of $y(\beta)$, $y'(\beta)$, to gradually approach 0 as $\beta$ approaches 90°. If there are no ice cliffs within the spatial domain, the iterative process will end as soon as $A_{cliff_i} = 0$ which will likely occur at a lower $\beta$ relative to spatial domain with ice cliffs because debris cover can only be maintained on a subset distribution of surface slopes (see inset histograms in Fig. 3). This truncation is likely the key distinction of areas with no ice cliffs relative to ice cliff abundant domains (see Sect. 5.2)."

```
P.8,L.16:  The formulation most accurate final A_cliff and coupled p(x) is
not necessarily true.  It is probably the optimum combination of A_cliff
and p(x).
```

Changed as suggested.

```
P.8,L.22:  please refer to Fig.  6 already here, so that the reader can
relate this difficult description to the graphical expression
```

Reference added.

```
P.8, eq.4:  what mathematical form should ''distance (P1,P2,(beta,
y(beta))) represent''?  This formulation seems strange.  Can you give
some indication how you came to the resulting right hand side of eq.  4?
```

Following "distance (P1,P2,(beta, y(beta))" we give the expanded form of this shorthand. We describe the shorthand in the text: "$d$ is the orthogonal distance from a line defined by points $P_1$ and $P_2$ to the function $y(\beta)$" We are not sure who to cite for the distance from a point to a line equation.

```
P.9;l.5:  It should already be mentioned here that gama has small
thresholds.
```

Changed as suggested: "'$\gamma$ is an input parameter with a near zero value (Table 1) defining...'

> P.9,L.6:  I do not see any problem with also calculating realistic
> values for P2 at beta=90°.  Maybe it would be a better approach to use
> the point of inflexion of the gamma function as optimum.  Because this
> is defined, based on the approximated parameters a, b and c only

Since the Reviewer cites parameters a, b, and c we assume she/he means the point of inflection of the function $y(\beta)$ (Eq. 2) rather than a gamma function. This was the initial approach we tried when designing the technique, however, the so-called 'elbow' point we were after can be found as an inflection only of a high order (4th) derivative which could be a mathematically unstable quantity. We therefore deemed it best to add an additional parameter with the hypothesis that it might be stable when applied in different regions on Earth.

> P9,L.10:  This is not an additional iteration, but a final application
> of the model with the found $\beta_{opt}$.

Changed as suggested.

> P9.,L.11/12:  Does this indicate that you just use a manual $\beta_i$ if the
> methods fails?  What would be the potential reasons for the method
> failing?

We added "manual[ly]" for clarity and added a sentence describing why we think the method could fail: "If visual inspection suggests large errors, all of the ice cliff probability maps and resulting ice cliff area shapefiles generated from the earlier set of iterations are retained and can be manually assessed to establish if a more adequate $\beta_i$ value should be considered optimal. Future applications of this method that produce large errors might indicate that a fixed $\gamma$ value is not suitable for all regions or that the surface slope distributions of debris covered area and ice cliff area have too much overlap to use surface slope alone as a deterministic attribute."

> P.9,L. 20:  The fractional debris covered area represents the original
> classification of debris covered glacier, where embedded clean ice is
> included in the debris cover?

"(considering only debris-covered area)" was added to the first sentence of this section: "Using this method over large spatial domains might be computationally demanding on typical desktop or laptop computers. To address this, a precursory function segments large domains (considering only debris-covered area) into less computationally taxing tiles."

> P.10,L.6:  The truth dataset x is probably different from the parameter
> x in p(x).  Can you explain?

It was intended to be the same, read as 'probability that a pixel falls within x given $\beta$ and $\omega$. But we agree this was unclear. We have restructured this notation in a way that we believe is more correct. x is now defined as a pixel in the spatial domain and we solve for the probability that x=Ice cliff, given $\beta$ and $\omega$. We have removed all reference to the truth dataset being called $x$ and describe it each time it is mentioned with words.

> P.10,L.15:  What is the secant of a slope pixel?  Here it seems that
> you calculate a volume (length by area) instead of an area (length by
> length).  Please reformulate.

We have rewritten the method description for finding true surface area and hope that is now more clear: "True surface area (as opposed to map view surface area) was calculated to present final results (Sect. 4 and 5.3) by multiplying DEM pixel area by a factor correcting for constant-sloped terrain $(\cos(\beta)^{-1}$, the secant of slope angle) for each pixel and finding the sum of slope corrected area for all ice cliff pixels or pixels within the entire spatial domain."

> P.10.L.19/20:  Again please simplify:  ''tool output vector shape
> defined by $A_{cliff}$'' means probably the same as ''$A_{cliff}$ vector shapes''.

Changed as suggested.

> P.11,L.12:  delete ''for error distribution <1''

Changed as suggested.

> P.12,L.2:  Can you comment here on the color coded distribution of
> ''true positive'' results in fig.  7?  It seems there is a clear
> distinction in dependence of beta.  What is the meaning of that?

We think that addressing this comment at P.13 L.12 is a location more on topic with the reviewers point. We have changed the sentence there to: "The figure shows a clear *true positive rate* dependence on slope illustrating the limitations of this method to detect ice cliffs where steep surface slopes were not sufficiently resolved in the data."

> P.12,L.9:  magnitude (typo)

Correction made.

> P.12,L.19:  What would be the difference in this example of Fig.  6,
> using $P_1$ ($beta_i$) for 60° and 90°?

We apologise but do not completely follow this question or what changes/corrections might be implied. If P1 is extended further from the position set by Eq. 5 the final ice cliff area would be less, in this way the method is sensitive to $\gamma$.

> P.12,L.20/21:  this needs more emphasis, that the truly artificial
> determination of $beta_{opt}$ from the Gaussian distribution matches with the
> error optimum based on the calibration.

Changed as suggested.

> P.13,L.14/15:  I do not understand this sentence.  Can you please
> elaborate on the limitations with respect to slope angle?

Changed to: "The figure shows a clear *true positive rate* dependence on slope illustrating the limitations of this method to detect ice cliffs where steep surface slopes were not sufficiently resolved in the data."

> P.13,L.14:  Low slope means that the spatial dimension of the cliff is
> so narrow that the elevation difference between pixels is dominated by
> the more flat debris covered parts.  The cliff portion itself is still
> "steep".  Beyond the ice cliff means, beyond the spatial classification
> result

This is what we meant by "...were not resolved in the data." We added the word " sufficiently" and removed mention of the possible condition of a low surface slope ice cliff, we agree this is somewhat oxymoronic.

> P.13,L.19:  What means ''surface slope may be saturated''?

Drawing from the Reviewers comment above we changed this sentence to:"The abundance of small ice cliffs with a very low *true positive rate* shown in Fig. 7 indicate improvements to automated ice cliff detection will need to, in part, focus on small ice cliffs where the elevation difference between pixels might be dominated by the more flat surrounding debris-covered area and cause a dampening the steep ice cliff signal."

> P.15,L.11:  The example of Figure 8 should be accompanied by the
> characteristics of the ice cliff results from Canwell Glacier, because
> the performance depends clearly on the typical ice cliff width and slope
> distribution.

Added: "With a coarsening of DEM resolution, larger ice cliffs were still correctly identified but with a loss of precision in ice cliff geometry and smaller ice cliffs dropped below the detection limit."

> P.15,L.17:  There still exists a result for $\beta_{opt}$:  about 19° and
> $y_{opt}$:  about 4%.  The only indication that the method fails is the
> non-assymptotic shape of the fuction towards the x-axis?  What is the
> reasoning behind this?

We had given this reasoning, following P.15,L.17: "A domain with ice cliffs will likely have at least a few very steep areas that will carry computations through higher values of $\beta_i$, while simple undulating surface topography should not produce abruptly steep slope values and as soon as $\beta_i$ exceeds the maximum slope present, iterations will stop. " but have rewritten this in an effort to be more precise: "While there is likely a range of slopes where debris-covered area and ice cliff area will both exists (inset histograms in Fig. 3), ice cliffs, by definition, are steep features that will carry iterations towards 90°. If this ice cliff component is not present,

the iterations will terminate as soon as there is no longer area with a slope above the slope threshold for a given iteration. This termination is a key characteristic that could indicate there are no ice cliffs within the spatial domain (Fig. 9)."

> P.15,L.29:  What is meant with ''mass wasting''?  Is it removal of
> supra-glacial debris?  Mass wasting usually is used for ice loss

"(geomorphic)" added before "mass wasting".

> Fig.  5:  Panel (e) is not mentioned on the caption.

Panel (e) description has been added.

---

## Author Response (AR2)

**Response to the Editor's comments regarding Herreid and Pellicciotti: "Automated detection of ice cliffs within supraglacial debris cover"**

https://doi.org/10.5194/tc-2017-151

April 12, 2018

**Comments from Tobias Bolch (Editor)**

Thank you, Tobias, for your thorough read of the manuscript and very constructive comments. All of your comments were adopted into the manuscript with a few small exceptions. Each comment is addressed in detail, point by point, below. Page and line numbers used are referenced to the same document that you reference in your comments (track change version from prior submission).

> In a strict sense you are using a digital surface model (DSM) of the
> glacier (surface). This term should at least be mentioned in the
> beginning. More and more higher resolution DEMs are becoming available,
> but for many parts of the Earth there is no DEM with a resolution of 5m
> or better available and I would also not (yet) see these high resolution
> as ''conventional''. If available for free, these DEMs with relatively
> high resolutions provide often a merge of several scenes and only one
> snapshot in time. However, cliffs are changing. In addition, they can
> contain significant errors as the data is processed for large areas
> automatically. This should be clarified and the term ''conventional''
> avoided for the DEMs needed. You may also mention the new HMA DEM with
> a resolution of 8m (Shean, 2017) and also the TanDEM-X DEM with 12m
> resolution.

To address this comment, we have (1) added to P4L18: "(specifically for the examples presented here, a digital surface model)"; (2) removed the term "conventional" throughout the manuscript in regard to input DEMs; and (3) expanded on P5L16-19 to now read: "DEM data that meet this criteria are not freely available for all glacierized regions on Earth at the present time. However, recent advances such as (1) the Interagency Arctic Research Policy Committee which has released Arctic DEM (http://ArcticDEM.org), a freely available 2 to 8 m resolution DEM for all landmass above 60° latitude and the entire State of Alaska; (2) a freely available 8 m resolution DEM for high mountain Asia (https://nsidc.org/data/highmountainasia); and (3) 12 m resolution TanDEM-X DEM data which may be freely available per a scientific data acquisition request (https://tandemx-science.dlr.de/) all show promise that high resolution DEM data in glacierized areas may soon be available globally."

**Specific comments**

> P4L7:  <1 m should be written in brackets

Changed as suggested.

> P4L13:  You cannot start a sentence with ''<1''.  Reformulate

Sentence changed to: "Visible imagery with a spatial resolution of less than 1 m was collected in the Alaska Range and used to assess the abundance of "thin" debris cover on ice cliff faces."

> P4L18:  What is .17ex 5 m resolution?  Should be clarified.  In addition,
> you should clarify here that one does not need a DEM with a resolution
> of 5m, but that the method works also for lower resolution DEMs.

Apologies, this is a snippet of LaTeX code used to draw a tilde without the use of additional packages. This was incidentally printed while compiling the track change version but is absent in the non-track change versions.

> L20:  But they should also not be too different in time.  Debris-cover
> is changing.

We think the Editor's comment is addressed in the following sentence L20-21: "However, debris-covered area and glacier margins should be assessed to ensure they have not changed significantly over the time span of the data used." But are happy to revise this sentence if it is insufficiently explicit.

> P5L4f:  Be specific; which sensor with which resolution did you use?
> Mention the advantage of S2 (10m res.  multispectral vs.  30m (15m pan).

We added further details to this sentence which now reads: "Data from the NASA/USGS Landsat program (used in this study, NIR: OLI Band 5 (30 m); SWIR: OLI Band 6 (30 m)) and ESA Sentinel-2(NIR: Band 8 (10 m); SWIR: Band 11 (20 m)) are two data sources that meet the input spectral and resolution requirements to map debris cover and are freely available."

> L12:  ''Structure from Motion'' is a method not data.  Clarify to which
> data you refer to.

Changed to: "This can simplify data processing if the input DEM is derived using structure from motion photogrammetry." We were uncertain about whether it is appropriate to capitalize 'structure' and 'motion'. Since we only use the term twice in the article we thought it would be easier to read without an acronym but then is less clear how to capitalize. If there is preference or convention we are of course happy to comply.

> L15: I agree but you do not need high absolute height accuracy with
> respect to the geoid.

Language corrected.

> L17ff: Provide a reference or URL for the Arctic DEM and mention here
> also the new HMA DEM and other suitable DEM sources which might be
> available for free for scientific purposes (such as the TanDEM-X DEM)

Changed as suggested and described in the main comment response above.

> P7L12 ''was'' identified. Write here which sensor was used (OLI?)

Both changed as suggested.

> P8L24: ''Eq. ??'' ?

Apologies again, this is an artifact from the LaTeX diff track changes version and is not present
in the final, non-track changes version.

> P15L12f. (and also elsewhere): I do not like sentences like'' Table 3
> shows...'' as this is a bit waste of space as Table/Figure captions are
> often similar. Write the most important finding and then refer to the
> table in brackets. However, this is a matter of style and my personal
> view, so feel free to leave these sentences as they are.

We have corrected this as suggested throughout the manuscript with two exceptions: we thought
P14L33 and P18L11, which both discuss fairly complex figures, were best described with this
style of language.

> P17L10 ''2x'' -> two times

Changed as suggested.

> L12ff: See my comment P15L12f, but more important: I find the
> comparison of the results based on different resolutions very relevant
> and interesting. This should be extended a bit more. If you could
> compare also to freely available DEMs for the same glacier(s) (HMA DEM
> vs. TanDEM-X, SRTM1, National DEMs etc. depending on coverage) for the
> same glacier to show the differences would from my point of view clearly
> increase the impact of the paper.

We are not sure what else we could say at P17L12-18 to extend this, but we added in the
Conclusion section "The DEMs used in this study had a spatial resolution of 5 m but, for

future applications of this method with different input DEM sources, we derived a relation between coarsening DEM resolution and method performance which can help guide anticipated outcomes." to further highlight this aspect of the study. We completely agree that a study of sensor error would improve the discussion of repeatability and the circumstances under which methods like this one can be used with confidence, but we think this deviates from the main goal of an ice cliff mapping method and deserves a more detailed investigation than what we would be able to do quickly now (e.g. even determining a maximum span of time between data acquisitions that is acceptable for such a study to be built on). A (planned) follow up study looking at wide-scale temporal evolution of ice cliffs would be a logical place to look for data with sufficiently close acquisition dates and derive sensor deviation/error.

> P18L3 ``geomorphic''

Changed as suggested.

> P20L13: ``Kääb''

Changed as suggested.

> Table and Figure captions: The captions are partly very long. It is
> again a matter of style but from my point of view long captions distract
> from reading the main text and with clear Figures (incl. legends) and
> Tables (incl. clear headings for the columns) long captions are not
> needed. Baseline information from the main text is also not needed (e.g.
> DEM resolution for this study was 5m).

Thank you for this good comment. We have added column headings to the tables which helped cut caption length substantially. We also applied this framework to the brand new Figure 1. We did not however, make very many changes to the remaining captions.

> Tables 2 and 3 could be shown in one. There is enough space as the
> letters of the tables are even large than of the text.

We have shrunk the table text to "footnotesize" in LaTeX and still we were unable to squeeze the two tables together. Perhaps if the typesetter engages at this level of detail they could help with a clever solution? Due to this we have left it as two tables (now with smaller letters) but only because we could not get it to work without making a sideways table which is probably less desirable.

> Figure 1: I find it hard to read. I would clearly prefer a usual map
> projection (maybe centred in eastern China or Japan). I cannot identify
> the location of the both glaciers on the map. I find also the oblique
> view not well done and the black lines hard to identify. I would use
> nadir view and do not cut most of the upper glacier part. The sentence
> ``In the oblique inset maps, orange lines are the glacier extent and the
> black polygons define the spatial domains used in this study'' can fully
> be avoided when including a legend.

Figure 1 has been completely redone with nadir view plots and a legend allowing a major reduction in the caption length.

> Figure 2: The letters are too large { waste of space and visually not really nice. Show the area of the subsets in Fig. 1.

Figure 2 has been shrunk so the text is not too large and so the figure can easily fit in a single column. Subset ares added to Fig. 1.

> Figure 3: What is the source of the optical data used?

Imagery source added.

> Fig. 4: In line with Rev. 1 I' also show Le on the other side.

Changed as suggested.

> Figure 5: In line with the reviewer I ask to flip the colour coding for the temperature. It needs to be consistent with the other figures and cold should be blue. In case you want a similar look you may flip the ice cliff probability.

Changed as suggested.

[revised manuscript text omitted]